# How synaptic strength, short-term plasticity, and input synchrony contribute to neuronal spike output

**Moritz O. Buchholz**[1☯], **Alexandra Gastone Guilabert**[1☯], **Benjamin Ehret**[1], **Gregor F. P. Schuhknecht**[1,2]*

**1** Institute of Neuroinformatics, University of Zürich and ETH Zürich, Zürich, Switzerland, **2** Department of Molecular and Cellular Biology, Harvard University, Cambridge, Massachusetts, United States of America

☯ These authors contributed equally to this work.
* gregor_schuhknecht@fas.harvard.edu

**Data Availability Statement:** All relevant data are within the paper and the computer code is available under: https://github.com/gschuhknecht/Buchholz_GastoneGuilabert_2023.git.

## Abstract

Neurons integrate from thousands of synapses whose strengths span an order of magnitude. Intriguingly, in mouse neocortex, the few 'strong' synapses are formed between similarly tuned cells, suggesting they determine spiking output. This raises the question of how other computational primitives, including 'background' activity from the many 'weak' synapses, short-term plasticity, and temporal factors contribute to spiking. We used paired recordings and extracellular stimulation experiments to map excitatory postsynaptic potential (EPSP) amplitudes and paired-pulse ratios of synaptic connections formed between pyramidal neurons in layer 2/3 (L2/3) of barrel cortex. While net short-term plasticity was weak, strong synaptic connections were exclusively depressing. Importantly, we found no evidence for clustering of synaptic properties on individual neurons. Instead, EPSPs and paired-pulse ratios of connections converging onto the same cells spanned the full range observed across L2/3, which critically constrains theoretical models of cortical filtering. To investigate how different computational primitives of synaptic information processing interact to shape spiking, we developed a computational model of a pyramidal neuron in the excitatory L2/3 circuitry, which was constrained by our experiments and published *in vivo* data. We found that strong synapses were substantially depressed during ongoing activation and their ability to evoke correlated spiking primarily depended on their high temporal synchrony and high firing rates observed *in vivo*. However, despite this depression, their larger EPSP amplitudes strongly amplified information transfer and responsiveness. Thus, our results contribute to a nuanced framework of how cortical neurons exploit synergies between temporal coding, synaptic properties, and noise to transform synaptic inputs into spikes.

## Author summary

Pyramidal neurons in neocortex receive thousands of synapses from the cells around them. Most synapses are 'weak', i.e., they have only a small depolarizing effect on the

**Funding:** This work was supported by funding of the Swiss National Science Foundation to GS (P2EZP3_188017 and P500PB_203130) and by the University of Zürich, Switzerland. The funders had no role in study design, data collection and analysis, decision to publish, or preparation of the manuscript.

**Competing interests:** The authors declare no competing interests.

postsynaptic neuron. Intriguingly, the few 'strong' synapses are predominantly formed between those neurons that become active together during sensory stimulation. This suggests that synaptic strength may determine who fires together in the brain. However, many other factors are known to contribute to computation, including spike correlations, firing rates, and synaptic background noise. Here, we first used electrophysiological experiments to show that strong synapses also depress the most during continuous activation, suggesting their strengths could be substantially reduced when neurons fire with high frequencies *in vivo*. To investigate this in more detail, we built a computational model of a pyramidal neuron. Our model suggests that the temporal correlation of presynaptic spike trains primarily determines which inputs can evoke action potentials in the postsynaptic neuron. When these correlated inputs also form the strongest synapses, as shown *in vivo*, the information transfer of the correlated inputs and the responsiveness of the postsynaptic neuron are further amplified. Our study contributes to a nuanced framework of how pyramidal cells exploit synergies between temporal coding, synaptic properties, and noise.

## Introduction

Cortical neurons compute spiking responses based on synaptic inputs from thousands of cells in the surrounding brain tissue. The strengths of these inputs span an order of magnitude and follow a lognormal distribution: while the majority of synaptic connections evoke small excitatory postsynaptic potentials (EPSPs), a small minority elicits comparably large EPSPs [1–6]. Intriguingly, in mouse primary visual cortex (V1), the 'strong' connections were found between those neurons that exhibited the most similar receptive field properties *in vivo* [6]. From these observations, a simple organizational principle of synaptic strength was proposed, in which the majority of the synaptic excitation necessary for action potential firing is provided by a small fraction of strong inputs, which determine the spike output of the postsynaptic cell [6]. The notion that synaptic strength is the primary determinant for the functional properties of neocortical circuits is attractive because it suggests that mapping the strongest connections in functional or structural analyses reveals the underlying functional organization of neocortical circuits. However, a more complex picture emerged from a recent study in ferret V1, where the response selectivity of neurons to visual stimulation was found to be determined by the cumulative weight of all co-active synapses, and could not simply be predicted from the tuning of synapses with large EPSPs [7].

Several other observations give further weight to the notion that synaptic strength alone is insufficient to explain neuronal response properties. Synapses are complex biophysical devices, whose response during ongoing activation is insufficiently captured by only a single weight parameter. It is intriguing that those synapses that elicit the largest EPSPs also tend to exhibit the most pronounced short-term depression [8–10], which can vastly reduce the total charge a synapse can deliver to its postsynaptic partner during repeated activation [11–16]. Thus, synaptic connections with large EPSPs recorded *in vitro* may operate in a substantially depressed state *in vivo* due to ongoing spontaneous and stimulus-evoked activation [15]. Furthermore, even the largest EPSP amplitudes provide only a fraction of the depolarizing charge necessary to drive the membrane potential of a cortical neuron through the spike threshold. Thus, temporal coincidence in presynaptic spike trains must necessarily be an important factor for information coding in neocortex [7,16–19]. Finally, neurons *in vivo* operate in the presence of continuous bombardment with synaptic background activity. Spontaneous firing rates of pyramidal cells in the superficial layers of rodent sensory areas range from 0.1 to 0.4 Hz *in vivo*

[20–25]. Because pyramidal neurons in rodent sensory areas are estimated to receive input from up to ~8000 synapses [26], they must experience hundreds to thousands of spontaneous synaptic events per second. In rodent V1, synaptic connections with small EPSPs occur predominantly between cells that display different response properties and thus fire with little temporal synchrony during visual stimulation [6]. Thus, in rodent sensory areas, the vast majority of excitatory synapses formed with any pyramidal neuron provide a constant bombardment of excitation that is relatively unrelated to the tuning of that neuron. Therefore, to compute spiking responses from their synaptic inputs, neocortical neurons operate in a complex parameter space. While much research has been conducted on the computational role of synaptic strength, e.g. [6,7,27], short-term plasticity, e.g. [12,13,17,28–30], and the temporal structure within synaptic inputs, e.g. [16–19], it remains much less studied how these parameters interact to shape information transfer in sensory areas.

Here, we combined experimental work and data-driven computational modeling to investigate systematically how this complex parameter space could shape the spiking responses of pyramidal neurons in L2/3 of mouse barrel cortex (S1). The distributions and patterns of action potential firing rates [21,23,25,31], synaptic strengths [6,27,32], correlations within neuronal activity [22,33], and temporal correlations within synaptic inputs converging onto the same neuron [6] have been well-characterized for L2/3 in rodent sensory areas *in vivo*. However, even though paired-pulse ratios have been measured for excitatory synapses across all cortical layers and different areas and species, most studies relied on small datasets that aimed to detect general differences in the mean [4,8–10,32,34]. Thus, a detailed characterization of the exact statistical distribution of short-term plasticity in mouse sensory L2/3 is missing. Likewise, the relationship between synaptic strength and short-term plasticity has not been characterized clearly for L2/3. Finally, it remains an open theory-inspired question whether synaptic connections that converge onto the same neuron exhibit a systematic bias of EPSP amplitudes [35] or short-term plasticity, which could endow individual neurons with low-pass filter or high-pass filter properties, if they were to receive predominantly depressing or facilitating synapses, respectively [14,36–38]. We addressed these questions by using two complementary methods to map synaptic transmission between L2/3 pyramidal neurons in barrel cortex slices: (1) minimal extracellular stimulation of putative single axons of passage in combination with whole-cell recordings of L2/3 pyramidal cells and (2) paired recordings of synaptically connected L2/3 pyramidal neurons. Then, we developed a computational model of a L2/3 pyramidal neuron that received excitatory inputs from 270 other L2/3 neurons [39], whose synaptic strengths and short-term plasticity were modeled after our experimental data. Presynaptic inputs were set to display temporal firing patterns constrained by *in vivo* data: the few synaptic connections eliciting large EPSPs fired temporally correlated spikes at high frequencies and were termed 'strong' inputs, while the more numerous connections triggering small EPSPs–termed 'weak' inputs–fired uncorrelated spikes at lower frequencies [6]. By selectively manipulating the relationship between synaptic strength, short-term plasticity, and temporal structure in the synaptic inputs, we characterized the importance of each of these parameters and their interdependencies in our simulation.

## Results

### Mapping synaptic strength and short-term plasticity in L2/3

We characterized the distribution of EPSP amplitudes and corresponding paired-pulse ratios of excitatory synaptic connections formed with regular-spiking neurons in L2/3 of barrel cortex and tested the theoretical prediction that synaptic connections converging onto the same postsynaptic cell may have systematically biased strength [35] or short-term plasticity

properties. While paired recordings are the gold standard for measuring synaptic transmission between identified neurons, they usually allow to record only one or few synaptic connections formed with the same cell. Therefore, to be able to characterize multiple, different synaptic connections formed with the same postsynaptic neuron, we measured somatic whole-cell responses to extracellular paired-pulse stimulation of putative single axons at multiple locations in the surrounding L2/3 (Fig 1A and 1B).

We obtained recordings from 20 regular-spiking neurons for which we identified a total of 74 sites at which minimal extracellular stimulation evoked EPSPs (mean of 3.7 synaptic connections per neuron). For a subset of these cells, we performed post-hoc biocytin histology to confirm that they were indeed pyramidal neurons (Fig 1A; see *Methods*). We applied stringent quality controls to ensure that we activated putative single axons of passage with our minimal stimulation protocol (see *Methods*). Briefly, we only included synaptic connections for which the smallest observable EPSP was evoked in an all-or-none manner in a fraction of trials and if the mean EPSP amplitude and failure rate remained constant throughout the recording [40,41]. To avoid recording the same axon repeatedly, different synaptic connections converging onto the same postsynaptic cell were only included if their location of stimulation was > 50 μm away from all previous stimulation locations.

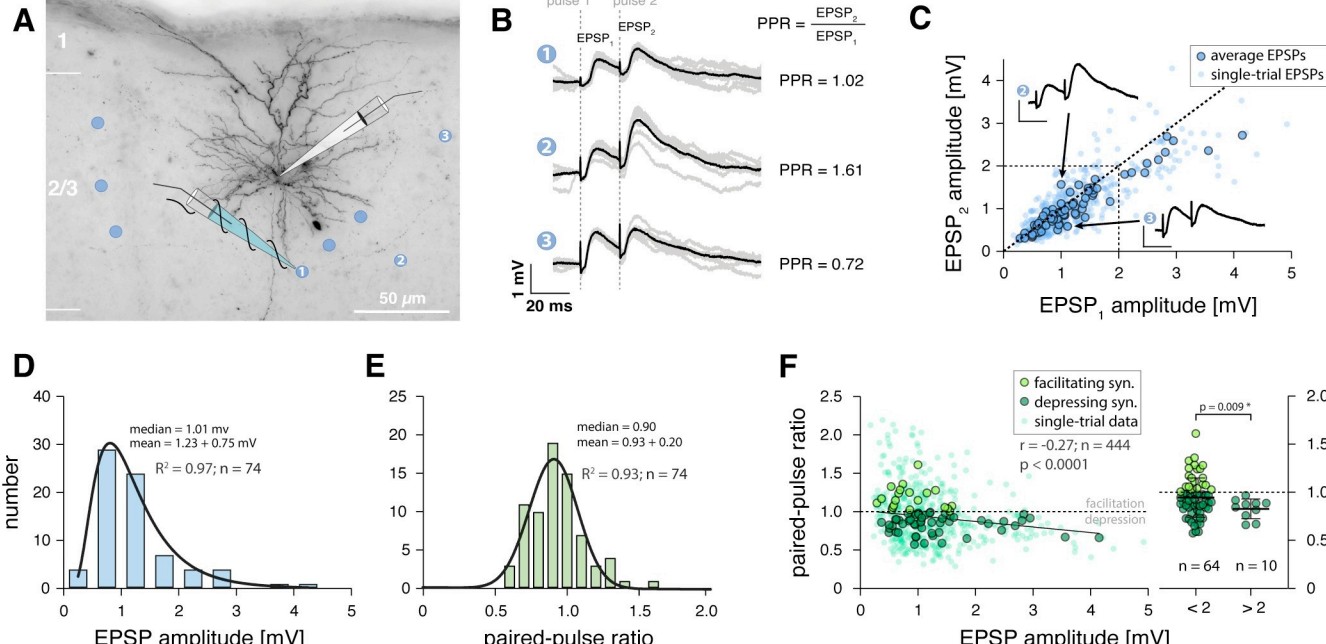

**Fig 1. EPSP amplitudes and paired-pulse ratios of excitatory synaptic connections formed with L2/3 pyramidal neurons in barrel cortex. A** Example of recorded regular-spiking L2/3 neuron in mouse barrel cortex visualized through post-hoc biocytin histology. Blue dots indicate locations of successful extracellular stimulation, blue pipette signifies extracellular stimulation electrode. The neuron's responses to stimulation at three different positions (labeled 1–3) are shown in B. **B** Somatic voltage recordings following 20 ms paired-pulse stimulation in the locations indicated by numbers. Grey traces, individual trials; black traces, average response; paired-pulse ratios (PPR) indicated. For timing of extracellular stimulation pulses (dashed lines), note the electrical stimulation artifact in somatic voltage responses. **C** Scatter plot showing, for all synaptic connections, the response to the second pulse versus the response to the first pulse (corresponding to the EPSP amplitude) of the paired-pulse stimulation paradigm. Large dots, average for each connection (n = 74); small dots, six randomly selected single-trial responses for each connection (n = 444). Data points below diagonal indicate depressing synaptic connections, dots above diagonal indicate facilitating connections. Voltage traces, same as traces 2 and 3 in B with identical scale bars. **D** Distribution of average EPSP amplitudes, histograms were fit with lognormal functions ($R^2$, goodness of fit). **E** Distribution of average paired-pulse ratios, histograms were fit with Gaussian functions ($R^2$, goodness of fit). **F** Left, scatter plot showing relationship between EPSP amplitude and paired-pulse ratio; large dots, average for each connection (n = 74; light green, facilitating connections; dark green, depressing connections); small dots, six randomly selected single-trial responses for each connection. The line was fit using linear regression, Pearson correlation results for single-trial data indicated. Right, comparison of paired-pulse ratios that were binned into 'weak' (EPSP < 2 mV) and 'strong' (EPSP > 2 mV) synaptic connections (parametric Welch's t test).

The distribution of peak amplitudes across the 74 EPSPs ranged from 0.29 mV to 4.15 mV (mean ± s.d.: 1.23 ± 0.75 mV), was markedly right skewed and could be fit well with a lognormal distribution ($R^2$ = 0.97) (Fig 1D). The mean coefficient of variation was 0.19 ± 0.06, the mean EPSP onset latency was 2.14 ± 1.12 ms and the mean 10–90% rise time was 2.54 ± 0.86 ms. For all 74 synaptic connections, we also recorded the paired-pulse ratio at an inter-spike interval of 20 ms (corresponding to a frequency of 50 Hz). Interestingly, the distribution of paired-pulse ratios appeared noticeably symmetrical with a mean ± s.d. of 0.93 ± 0.20 and could be fit well with a normal distribution ($R^2$ = 0.93) (Fig 1E). We found no correlation between EPSP amplitude or paired-pulse ratio and the age of the animal [8] (S1 Fig).

Because EPSP amplitudes followed a lognormal distribution, while their corresponding paired-pulse ratios were normally distributed, the question arose of how they could be mapped onto one another, i.e., whether there was a systematic relationship between synaptic strength and short-term plasticity. Interestingly, a scatter plot of the response amplitudes to the 2nd stimulation pulse against the response amplitudes to the 1st stimulation pulse (corresponding to the EPSP amplitude) showed the tendency that synaptic connections with larger EPSPs were depressing, while connections with smaller EPSPs exhibited a range of facilitating and depressing paired-pulse ratios (Fig 1C and 1F). While there was no significant correlation in our dataset between average EPSP amplitude and short-term plasticity, a negative correlation emerged when we plotted EPSP amplitudes and corresponding short-term plasticity on a trial-to-trial basis (Fig 1F).

To investigate this question further, we binned our dataset of synaptic connections depending on their average EPSP amplitude (into 0.5 mV bins, S1 Data). Critically, we found that in all bins with EPSP amplitudes below 2 mV, synaptic connections displayed a range of facilitating and depressing paired-pulse ratios. By contrast, all connections with EPSP amplitudes above 2 mV were depressing (n = 10) (Fig 1F). When we split the dataset accordingly, we found that connections below 2 mV had a mean paired-pulse ratio of 0.95 ± 0.20 (i.e., exhibiting little net short-term plasticity), while connections above 2 mV had a lower mean paired-pulse ratio of 0.83 ± 0.10 (Fig 1F).

## No clustering of connections with similar synaptic properties on individual postsynaptic neurons

Next, we investigated the open question of whether EPSP amplitudes and short-term plasticity across those synaptic connections formed with the same postsynaptic neurons followed the same distributions as those of all 74 connections across all regular-spiking neurons. Alternatively, the synaptic inputs onto a given cortical neuron may be statistically correlated, i.e., individual neurons could receive synaptic connections with systematically biased EPSP amplitudes or paired-pulse ratios that deviate from the overall distributions found across L2/3, which may constitute a mechanism to endow individual cells with high-pass or low-pass filtering properties [14,38]. For a total of 8 neurons, we were able to characterize at least 5 different afferent synaptic connections (47 connections in total, mean of 5.9 connections per cell). We will refer to the distribution of paired-pulse ratios and EPSP amplitudes across all our recorded synapses as the "population distribution" and to the distributions of paired-pulse ratios and EPSPs of synaptic connections converging onto a single cell as "cell distributions". We used the non-parametric Kolmogorov-Smirnov test to detect if there was a significant difference between the respective cell distributions and the population distribution. Interestingly, for all 8 cells, the cell distributions were not significantly different from the population distribution for both EPSP amplitudes and paired-pulse ratios (Fig 2A and 2B). We did not correct these results for multiple comparisons; instead, we conducted a power analysis to estimate the detectable effect size across the experimental series, which accounted for our multiple-testing strategy.

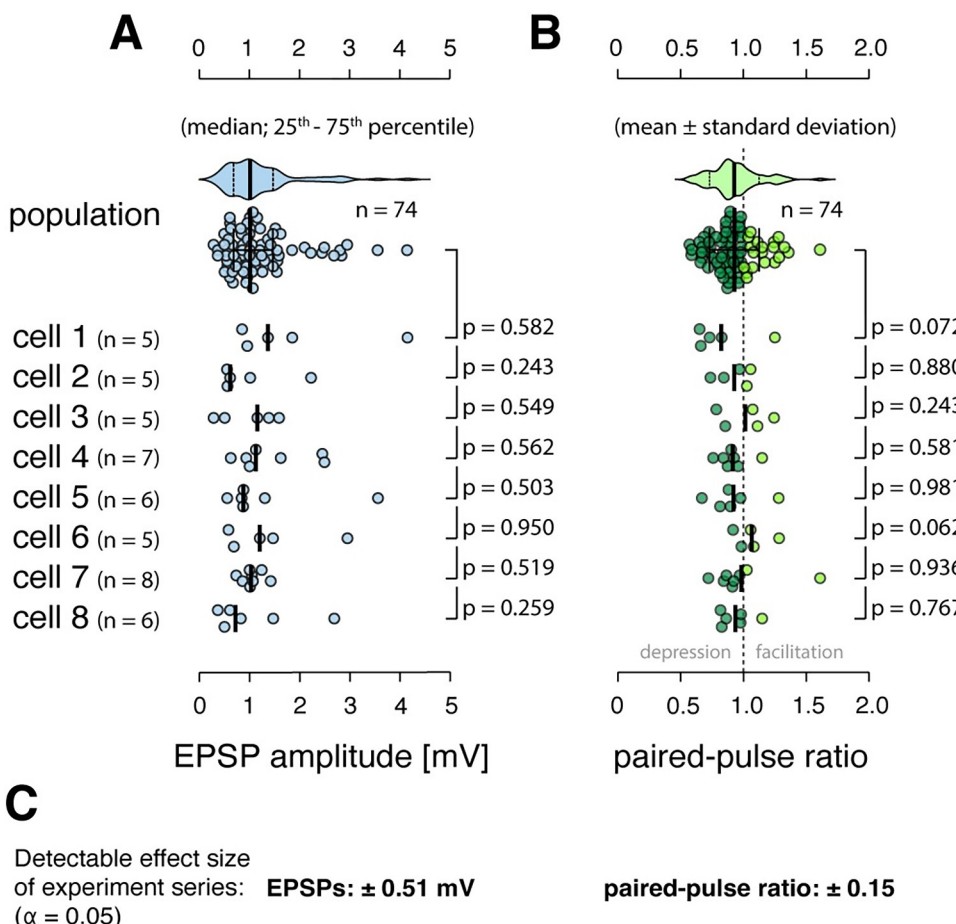

**Fig 2. Excitatory synaptic connections formed with L2/3 neurons do not exhibit a systematic clustering of EPSP amplitude and short-term plasticity. A** Top, distribution of EPSP amplitudes recorded across all regular-spiking neurons (population distribution). Bottom, distributions of EPSP amplitudes measured across the 8 neurons, for which at least 5 synapses were found (cell distributions). N, number of synapses recorded per cell; p, non-parametric Kolmogorov-Smirnov test between each cell distribution and the population distribution, medians are indicated. Note that the synapses formed with the postsynaptic cell in any given experiment were removed from the respective population distribution that the cell was compared to. **B** Same analyses for short-term plasticity data, panel layout as in A; light green, facilitating synaptic connections; dark green, depressing connections, means are indicated. Note that the synapses formed with the postsynaptic cell in any given experiment were removed from the respective population distribution that the cell was compared to. **C** Estimation of the effect sizes that were detectable across the experimental series at the 5% significance level.

Precise quantification of synaptic short-term plasticity requires electrophysiological recordings. Using whole-cell patch-clamp recordings in combination with minimal stimulation of axons of passage, however, limits the number of synaptic connections that can be recorded for any given neuron, yielding low statistical power on the level of individual cells. Therefore, we conducted a power analysis to estimate the detectable effect sizes in our dataset (see *Methods* for details). For detecting a significant ($\alpha = 0.05$) difference between each of the 8 paired-pulse ratio cell distributions and the population distribution, the Kolmogorov-Smirnov test had an average power of 17% for an effect size of 0.1, a power of 53% for an effect size of 0.2, and a power of 85% for an effect size of 0.3, where effect size corresponds to a systematic difference in the means of the cell distributions. Thus, the statistical power was low on the level of individual experiments. Because we could repeat the experiment 8 times, however, even small

systematic differences between cell distributions and population distribution, while undetectable in single experiments, should have been revealed in at least one or a few of the 8 neurons we recorded from.

To investigate this further, we used a binomial model (see *Methods*) to assess the power of the entire experimental series by asking: what systematic difference in paired-pulse ratios should have been observed in at least one of the 8 experiments at the 95% significance level? We found that the probability to detect a significant difference across our entire dataset was 78% for an effect size of 0.1 and 99.7% for an effect size of 0.2, with the 95% significance level at an effect size of 0.15. Critically, an effect size of 0.15 is below the paired-pulse ratio difference of 0.16 that we detected between the small- and large-EPSP connections in L2/3 (Fig 2C). Thus, our experimental series achieved the statistical power necessary to detect differences in paired-pulse ratios at physiological magnitudes that we found to exist in L2/3. This suggests that short-term plasticity of excitatory synapses formed with individual L2/3 pyramidal neurons spans the full range observed in L2/3 and is not markedly functionally clustered on the level of single cells.

Likewise, for detecting a significant difference between each of the 8 EPSP cell distributions and the population distribution, the Kolmogorov-Smirnov test had an average power of 4.9% for an effect size of 0.2 mV, a power of 15% for an effect size of 0.4 mV, and a power of 46% for an effect size of 0.6 mV. Analogous Monte Carlo simulations showed that the probability of detecting a significant difference in the mean EPSP amplitudes across our entire dataset was 72% for a systematic effect size of 0.4 mV and 99.3% for a systematic effect size of 0.6 mV, with the 95% significance level at 0.52 mV (Fig 2C). In summary, these are important experimental results that are inconsistent with the theory-inspired hypothesis that synaptic inputs onto single cortical neurons may be statistically correlated [35].

## Mapping synaptic transmission with paired whole-cell recordings

While minimal stimulation of axons of passage gave us the critical advantage to be able to map multiple synaptic connections formed with the same postsynaptic neurons, the method has several technical limitations that could potentially produce biased EPSP and paired-pulse ratio distributions. This includes that the origin of the stimulated axon remains unknown and could lie outside of L2/3, multiple axons of passage may be stimulated if the stimulation strength is not precisely controlled, action potentials may be evoked less reliably compared to paired recordings, different inputs may not be independent, and inhibitory axons could potentially be activated alongside the excitatory axon. To cross-check whether the synaptic properties we measured with extracellular stimulation were indeed representative of excitatory synaptic connections between L2/3 neurons, we additionally performed simultaneous whole-cell recordings of 22 pairs of synaptically connected pyramidal cells in L2/3 of mouse barrel cortex (Fig 3). Reassuringly, we could reproduce all results we had previously obtained with minimal stimulation experiments: EPSP amplitudes measured with paired recordings followed a lognormal distribution, while paired-pulse ratios were normally distributed (Fig 3C), and these distributions were not different from those measured with minimal simulation experiments (Fig 3D). As with our minimal stimulation data, we found a negative correlation between EPSP amplitude and paired-pulse ratios on the single-trial level (Fig 3E).

Finally, we tested whether our paired-pulse ratio measurements were representative of the short-term dynamics of L2/3 synapses during ongoing synaptic stimulation. Alternatively, a synapse that depresses between the first two pulses from rest may show facilitation during continuous activation, which has been demonstrated for example in the thalamocortical connection to mouse barrel cortex [30]. Therefore, in a subset of paired recordings, we activated

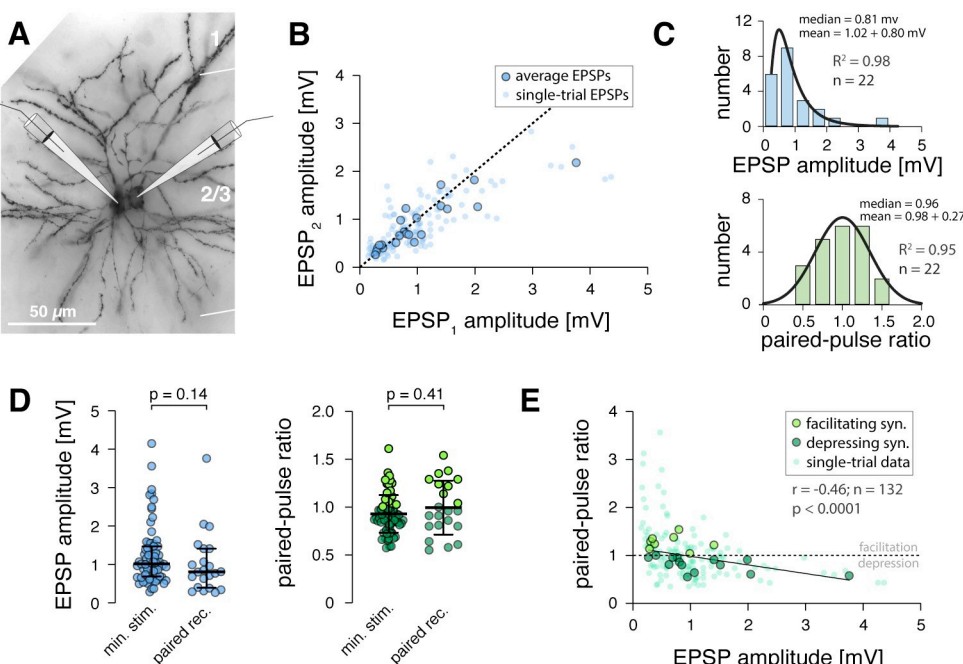

**Fig 3. Synaptic transmission between identified L2/3 pyramidal neurons measured with paired whole-cell recordings. A** Example of a synaptically connected pair of L2/3 pyramidal neurons in mouse barrel cortex visualized through post-hoc biocytin histology. **B** Scatter plot showing, for all synaptic connections, the response to the second pulse versus the response to the first pulse (corresponding to the EPSP amplitude) of the paired-pulse stimulation paradigm. Large dots, average for each connection (n = 22); small dots, six randomly selected single-trial responses for each connection (n = 132). Data points below diagonal indicate depressing synaptic connections, dots above diagonal indicate facilitating connections. **C** Top, distribution of average EPSP amplitudes recorded between connected L2/3 neurons, the histogram was fit with a lognormal function. Bottom, distribution of average paired-pulse ratios recorded between L2/3 neurons, the histogram was fit with a Gaussian function; $R^2$, goodness of fit. **D** Left, comparison of average EPSP amplitudes recorded with minimal stimulation (n = 74; same as in Figs 1D and 2A) and with paired recordings (n = 22). Right, comparison of average paired-pulse ratios recorded with minimal stimulation (n = 74; same as in Figs 1E and 2B) and with paired recordings (n = 22). Non-parametric Kolmogorov-Smirnov p-values are indicated. **E** Scatter plot showing relationship of EPSP amplitude and paired-pulse ratio for synaptic connections between L2/3 pyramidal neurons obtained with paired recordings. Large dots, average for each connection (n = 22; light green, facilitating connections; dark green, depressing connections); small dots, six randomly selected single-trial responses for each connection. The line was fit using linear regression, Pearson correlation results for single-trial data indicated.

synapses with a train of four action potentials at 50 Hz, which should suffice to reveal such variable short-term plasticity [30]. All synapses tested showed paired-pulse depression between the first two pulses and, interestingly, they continued to depress further during ongoing stimulation (S2 Fig). This suggests that the variable short-term dynamics observed in thalamocortical afferents may be a specific adaptation, e.g., to higher firing rates in the thalamocortical pathway, while short-term plasticity of synaptic connections between L2/3 pyramidal neurons behaves more uniformly during continuous activation.

## Modeling the interplay of synaptic strength, short-term plasticity, and input synchrony

We generated a leaky integrate-and-fire model of a L2/3 pyramidal neuron to investigate how synaptic strength, short-term plasticity, and temporal structure in synaptic inputs interact within the excitatory L2/3 circuitry to shape the response properties of cortical neurons (Fig 4A–4C; see *Methods* for details). For this purpose, we developed a data-driven modeling

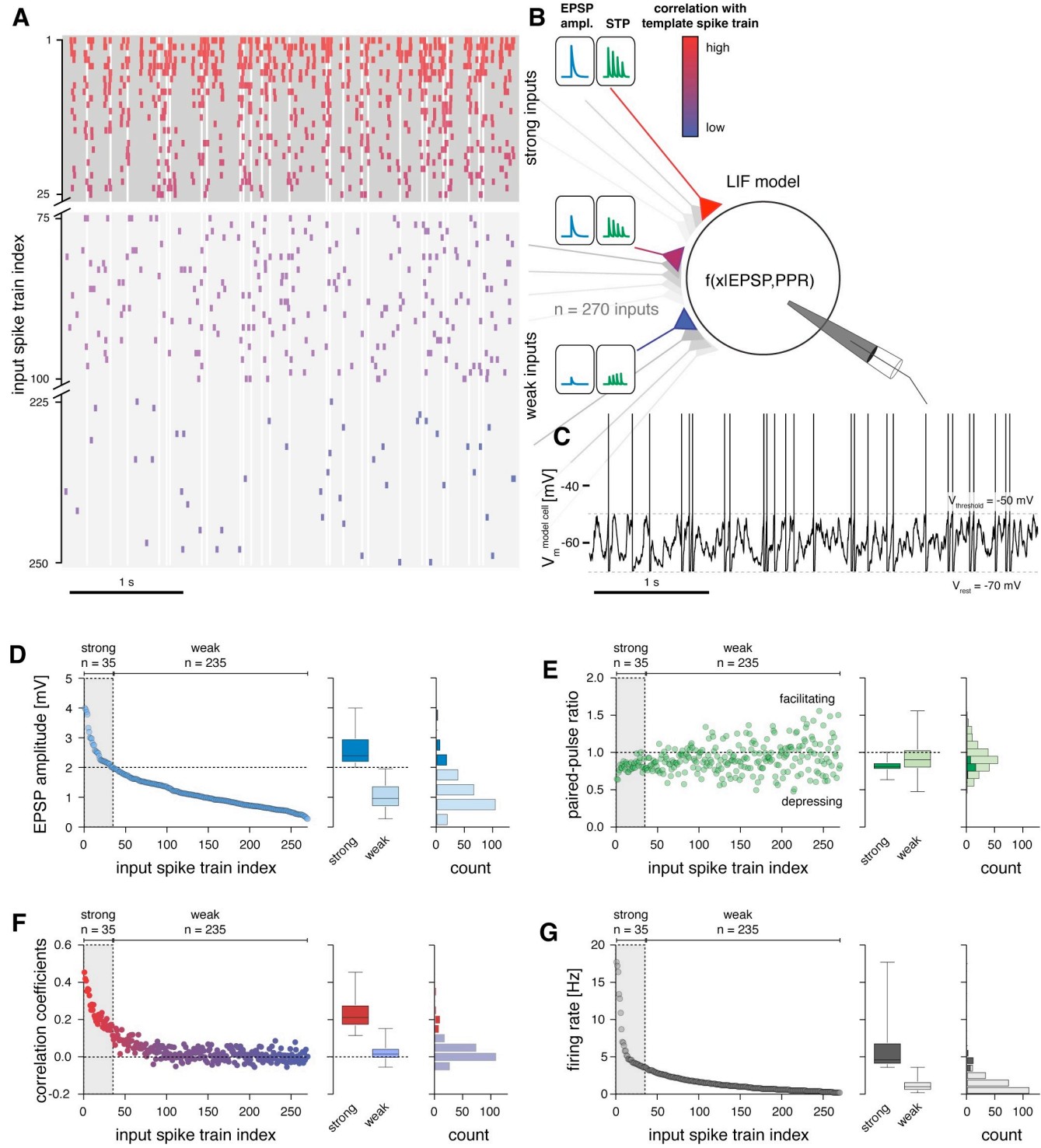

**Fig 4. Default setup of the L2/3 leaky integrate-and-fire neuron model. A** Example of input spike trains fed to the model cell. Strong inputs (top) fired with higher frequencies and temporal correlation (color coded) compared to weak inputs (bottom). Vertical grey bands indicate the resulting spike timing in the model cell (same as in C). Some weak inputs did not spike in the depicted 200 ms time window because of their low firing rates. **B** Strong inputs were set to have larger EPSP amplitudes and corresponding short-term depression, while weak inputs were set to evoke smaller EPSPs with weak net short-term plasticity, in accordance with our *in vitro* recordings. **C** Simulated membrane potential of model neuron following activation with the input spike trains shown in A. **D** Left, EPSP amplitudes across the 270 input spike trains. Center, comparison of EPSP amplitudes of strong and weak inputs (median, 25–75% percentile, and ranges are indicated). Right, same data plotted as histogram. **E** Left, 20 ms paired-pulse ratios across the 270 input spike trains. Center, comparison of paired-pulse ratios of strong and weak inputs (median, 25–75% percentile, and ranges are indicated). Right, same data plotted as histogram. **F**

Left, Pearson correlation coefficient between the 270 input spike trains and the template spike train that was used to generate the pairwise correlation structure (see *Methods*); color code as in A, B. Center, comparison of correlation of strong and weak inputs with template spike train (median, 25–75% percentile, and ranges are indicated). Right, same data plotted as histogram. **G** Left, firing rates of the 270 input spike trains. Center, comparison of firing rates of strong and weak inputs (median, 25–75% percentile, and ranges are indicated). Right, same data plotted as histogram.

approach: we constrained firing rates and pairwise correlations of presynaptic inputs by *in vivo* observations and synaptic strengths and short-term plasticity properties by our *in vitro* experimental data obtained with minimal stimulation.

Briefly, the model neuron received excitatory inputs from 270 presynaptic neurons [4,39], whose EPSP amplitudes (Fig 4D) and short-term plasticity properties (Fig 4E) were constrained following our extracellular stimulation experiments (see *Methods*). Note that this number of presynaptic L2/3 cells is based on the assumption that L2/3 neurons form on average 3 anatomical synapses with their postsynaptic partners in L2/3 [4,39]. In our model, this is captured by the fact that the axons of passage we activated with minimal stimulation must have also formed multiple anatomical synapses with the recorded neurons on average. This is evident when comparing the range of EPSP amplitudes we recorded with minimal stimulation (0.29–4.15 mV) with EPSP amplitudes obtained from paired recordings (0.15–2.25 mV) for which the number of anatomical synapses per connection (mean of 1.6) was additionally established from EM [42]. The temporal input correlations [6] (Fig 4F) and firing rates [25] (Fig 4G) across the 270 synaptic inputs were constrained by published *in vivo* data for rodent cortex (see *Methods*), such that a small number of strong synaptic inputs fired temporally correlated spikes at high frequencies and exhibited large EPSP amplitudes and corresponding short-term depression. The remaining majority of weak synapses, providing 'background' activity, were set to fire at low frequencies and in a temporally uncorrelated pattern, resembling a random Poisson process, and exhibited small EPSP amplitudes without pronounced net short-term plasticity (Fig 4A and 4B).

After the model was set up in this manner, we verified that all parameters were distributed following experimental data and that the interdependencies between EPSP amplitude and short-term plasticity and EPSP amplitude and temporal correlation structure [6] were preserved (S3 Fig).

To examine information transfer between the synaptic inputs and the output firing pattern of the model neuron, we measured the Pearson correlation coefficient between each input spike train and the model neuron's output spike train. We further characterized the neuronal gain of the model cell by mapping its input-output relationship (i.e., the probability of spiking as a function of the number of coincident synaptic inputs). By selectively manipulating the relationship between synaptic strength, short-term plasticity, and temporal structure in the synaptic inputs, we then systematically characterized the contribution of each of these parameters on information transfer and neuronal gain. Each experiment was repeated for a total of 100 simulation runs; for each iteration, we randomly re-generated a new set of 270 input spike trains.

Finally, because the model's spike output is directly determined by the specific set of model parameters, we assessed the reliability of our results by conducting a detailed robustness analysis. Briefly, we varied individual model parameters or combinations of parameters within ranges that have been experimentally reported for L2/3 pyramidal neurons *in vitro* and *in vivo* (see *Methods*). We found that while the absolute firing rate of the model depended on the specific parameter settings, the qualitative results of our correlation and gain analyses were relatively unaffected across the range of tested parameter combinations, under the constraint that the model neuron was allowed to spike at a sufficient firing rate (i.e., > 2 Hz) in its default setup to enable the correlation and gain analyses (S4 Fig).

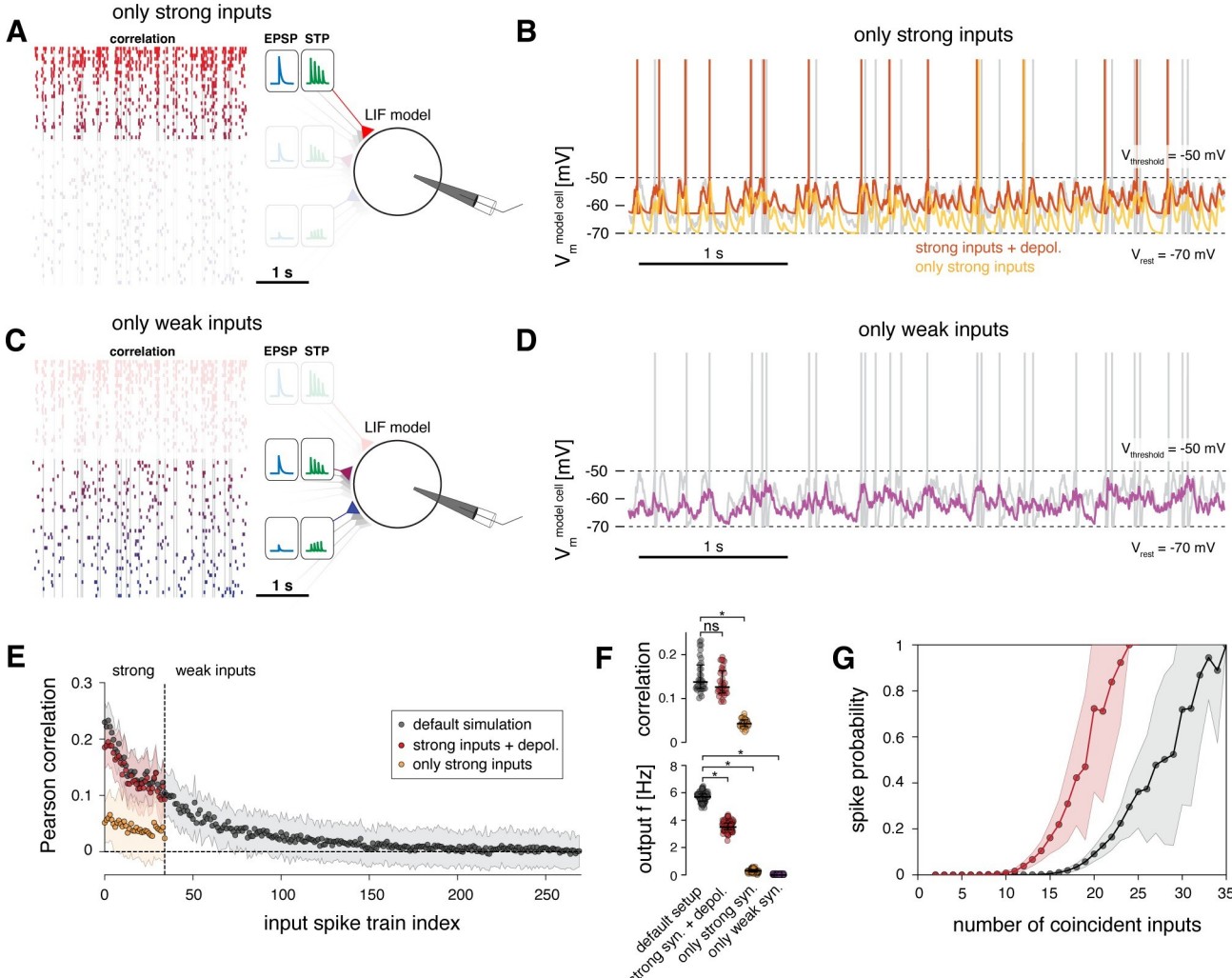

**Fig 5. Uncorrelated activity of weak inputs enhances information transfer of strong synaptic inputs. A** Schematic of model setup with weak inputs removed. **B** Example spike train of the model cell in its default setup (grey), when weak inputs are entirely removed (orange), and when weak inputs are replaced by a more depolarized $V_{rest}$ (red). **C** Schematic of model setup with strong inputs removed. **D** Example spike train of the model cell in its default setup (grey) and when strong inputs are removed (purple). **E** Pearson correlation coefficients of the 270 input spike trains with the output spike train of the model cell. Results of three model setups are shown: default simulation (grey; all inputs, as in Fig 4) and setups introduced in A (orange, weak inputs removed; red, weak inputs replaced with depolarized $V_{rest}$). Dots indicate means, shaded regions indicate standard deviation of correlation coefficients for 100 runs of the simulation. **F** Top, Pearson correlation coefficients between the strong synaptic inputs and the output spike train of the model neuron for the default simulation and setups introduced in A. Bottom, output firing rate of model cell for the default simulation and the setups introduced in A-C. (Data are averages across 100 simulation runs; median and 25–75% percentile indicated; non-parametric Kolmogorov-Smirnov test, * p < 0.05.) **G** Probability of output spiking as a function of the number of coincident spikes across all input spike trains (grey, default simulation; red, weak inputs replaced with depolarized $V_{rest}$).

First, we ran the simulation in its default 'physiological' setup, i.e., with parameters and parameter mappings as found in our *in vitro* recordings and published *in vivo* data (Figs 4 and 5, see *Methods*). For this, we set the resting membrane potential ($V_{rest}$) to -70 mV, which resulted in a mean output firing rate of 5.7 ± 0.3 Hz (Figs 4C, and 5F), in accordance with experimental measurements of *in vivo* spike rates in L2/3 of mouse barrel cortex [25]. In this setup, the model's average membrane potential ($V_m$) during synaptic bombardment by the weak and strong inputs was -59.8 ± 0.2 mV (Figs 4C, 5B and 5D), which is similar to *in vivo* whole-cell recordings in mouse L2 [9]. As expected, the strong synaptic inputs shared the highest Pearson correlation coefficients (mean ± s.d.: 0.15 ± 0.04; range: 0.11 to 0.23) with the

model's output spike train [6], while the weak inputs displayed correlation coefficients one order of magnitude smaller (mean ± s.d.: 0.02 ± 0.02; range: 0 to 0.10) (Fig 5E). Across all inputs, spike trains with decreasing intrinsic correlation, smaller EPSP amplitudes, and lower spike rates displayed increasingly lower correlation coefficients with the output spike train (Fig 5E). We confirmed that the Pearson correlation coefficients indeed detected correlations in spike timing rather than in firing rates by randomizing the output spike times following a random Poisson process while keeping the output firing rate identical. Reassuringly, the correlations between all inputs and the output spike train then dropped to zero.

## Synaptic background activity enhances information transfer of strong inputs

We probed the relative influence of the strong versus weak synaptic inputs on the output spiking of our model cell. Critically, when we removed the weak inputs entirely (Fig 5A and 5B), the mean correlation between the strong inputs and the output spike train was reduced to 0.04 ± 0.01 (Fig 5E and 5F). The output firing rate of the model neuron dropped to 0.29 ± 0.12 Hz (Fig 5F) and its average $V_m$ was hyperpolarized to -63.8 mV ± 0.07 mV. Despite this sharp drop in information transfer of the strong synaptic inputs, those inputs with the highest intrinsic correlation and synaptic strength still maintained the highest correlation with output spiking (Fig 5E).

To investigate whether, in our simulation, synaptic background activity enhanced strong inputs simply by depolarizing the membrane potential closer to the spiking threshold or by stochastic membrane fluctuations, we exchanged the weak inputs with a depolarized $V_{rest}$ (equivalent to the median $V_m$ measured when only weak inputs were active). We found that in the absence of stochastic membrane fluctuations, the correlation coefficients of the strong inputs were not different to the default setup; however, the overall responsiveness of the model neuron remained reduced (Fig 5F) [43]. Replacing weak inputs with a depolarized $V_{rest}$ also resulted in a steeper slope of the input-output curve (Fig 5G), confirming that synaptic 'background noise' has a divisive effect on neuronal gain. This noise broadens a neuron's sensitivity to the range of temporal correlations in input spike trains by increasing the time window over which coincident inputs can be integrated to evoke spiking, a finding in agreement with previous studies [44].

When we removed the strong synaptic inputs from the simulation (Fig 5C), uncorrelated activity provided by the weak inputs was by itself unable to drive the postsynaptic neuron above spiking threshold (Fig 5F). This is because the 235 weak inputs fired at an average frequency of 1.2 ± 0.9 Hz with mean EPSP amplitudes of 1.03 ± 0.42 mV, which resulted in a mean $V_m$ of -62.4 mV that rarely crossed the spike threshold (Fig 5D). Thus, uncorrelated activity of weak synapses alone was incapable of evoking spikes and did not transfer information encoded in its own spike trains (Fig 5E).

## Output spiking requires correlation and high firing rates of strong inputs

Next, we decoupled the high temporal correlation and high firing rates of strong inputs from their larger synaptic strengths by randomly assigning the EPSP amplitudes and their corresponding short-term plasticity properties across the input spike trains (Fig 6A). The original coupling between EPSP amplitude and short-term plasticity was maintained in this experiment, i.e., synapses with larger EPSPs still exhibited depression and synapses with smaller EPSPs exhibited facilitation.

When the model was set up in this manner, the firing rate of the output neuron decreased to 1.3 ± 0.3 Hz (Fig 6B and 6D). Critically, inputs with higher temporal correlation and higher firing rates still contributed more strongly to the firing of the model neuron (Pearson correlation mean ± s.d.: 0.08 ± 0.02) compared to inputs with lower temporal correlations and lower

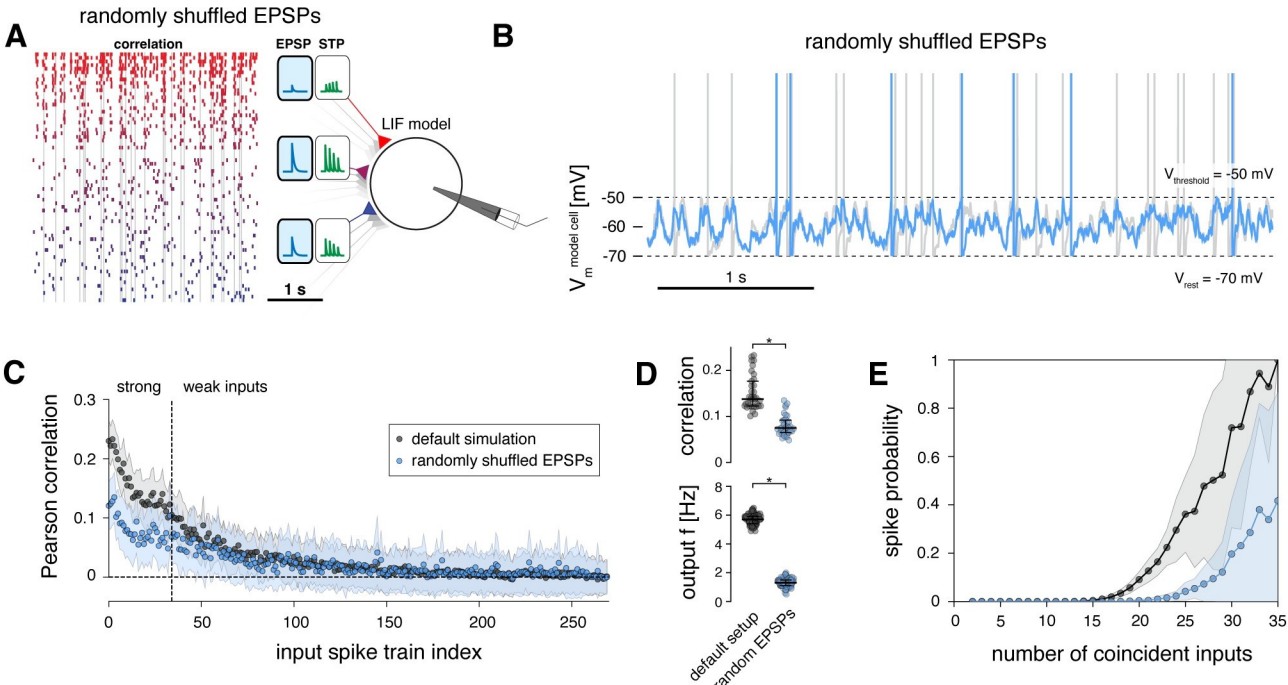

**Fig 6. Temporal correlation and firing rates primarily determine output spiking, synapse strength enhances responsiveness. A** Schematic of model setup with shuffled EPSP amplitudes; the relationship of EPSP amplitude and short-term plasticity was maintained**. B** Example spike train of the model cell in its default setup (grey) and with shuffled EPSP amplitudes (blue). **C** Pearson correlation coefficients of the 270 input spike trains with the output spike train of the model cell. Results of two model setups are shown: default simulation (as in Fig 4) and setup introduced in A. Dots indicate means, shaded regions indicate standard deviation of correlation coefficients for 100 runs of the simulation. **D** Top, Pearson correlation coefficients between the strong synaptic inputs and the output spike train of the model neuron for the default simulation and setup introduced in A. Bottom, output firing rate of model cell for the default simulation and the setup introduced in A. (Data are averages across 100 simulation runs; median and 25–75% percentile indicated; non-parametric Kolmogorov-Smirnov test, * p < 0.05.) **E** Probability of output spiking as a function of the number of coincident spikes across all input spike trains (grey, default simulation; blue, model setup with shuffled EPSPs).

firing rates (mean ± s.d.: 0.02 ± 0.02) (Fig 6C). This means that synaptic strength by itself did not determine which inputs transmitted the most information to the spike train of the output neuron. Instead, in our simulation, the combination of high temporal correlation and elevated firing rates was the primary determinant for evoking correlated spiking in the output neuron. However, matching larger EPSP amplitudes to inputs that fired with high temporal correlation and high firing rates (i.e., our default setup), as observed for strong synaptic inputs *in vivo* [6], further increased their correlation with the spike train of the model neuron by a factor of 2 and markedly enhanced the responsiveness of the model cell (Fig 6C and 6D). Decoupling the large EPSP amplitudes from the correlated inputs (by shuffling EPSP amplitudes amongst all input spike trains) also resulted in a flatter slope of the model's input-output curve and a reduced responsiveness to coincident inputs (Fig 6E). This suggests that assigning the largest EPSP amplitudes to those inputs that fire with high temporal correlation has a multiplicative effect on neuronal gain, leading to signal amplification as a mechanism to increase efficient information transmission of strong inputs [44].

## Short-term plasticity balances the computational effects of strong and weak inputs

Next, we removed the short-term plasticity mechanism from all synapses, such that they exhibited paired-pulse ratios of 1 for all inter-spike-interval durations, i.e., all EPSP amplitudes remained static during repeated stimulation (Fig 7A and 7B).

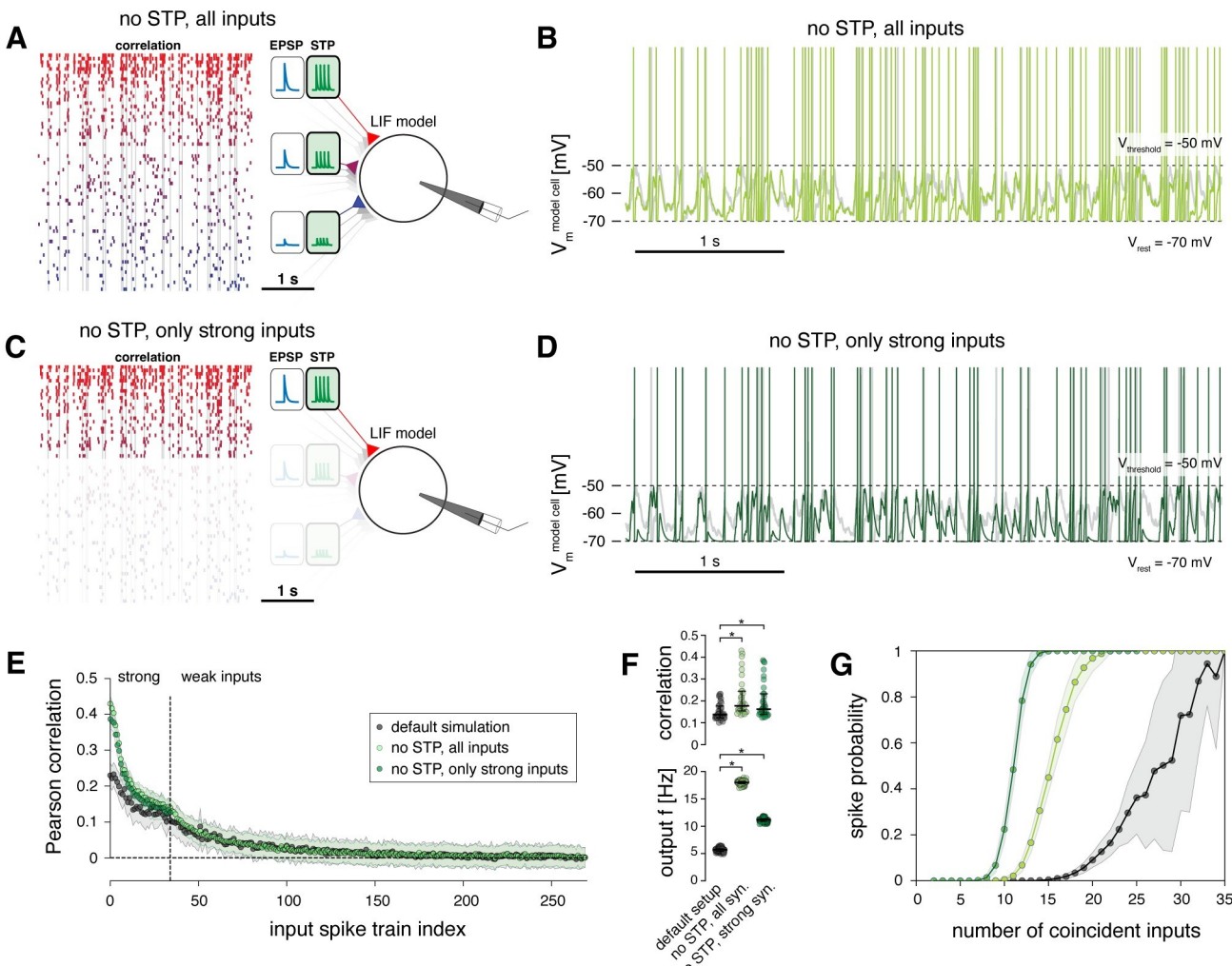

**Fig 7. Short-term plasticity balances the computational effects of strong and weak inputs. A** Schematic of model setup with short-term plasticity mechanisms removed, i.e., all spike trains exhibit a paired-pulse ratio of 1. **B** Example spike train of the model cell in its default setup (grey) and when short-term plasticity mechanisms are removed (light green). **C** Schematic of model setup with short-term plasticity mechanisms removed and the weak inputs removed in addition. **D** Example spike train of the model cell in its default setup (grey) and when short-term plasticity mechanisms and weak inputs are removed (dark green). **E** Pearson correlation coefficients of the 270 input spike trains with the output spike train of the model cell. Results of three model setups are shown: default simulation (as in Fig 4) and setups introduced in A, C. Dots indicate means, shaded regions indicate standard deviation of correlation coefficients for 100 runs of the simulation. **F** Top, Pearson correlation coefficients between the strong synaptic inputs and the output spike train of the model neuron for the default simulation and setups introduced in A-C. Bottom, output firing rate of model cell for the default simulation and the setups introduced in A-C. (Data are averages across 100 simulation runs; median and 25–75% percentile indicated; non-parametric Kolmogorov-Smirnov test, * p < 0.05.) **G** Probability of output spiking as a function of the number of coincident spikes across all input spike trains for the default simulation (grey) and the setups introduced in A-C (light green and dark green, respectively).

When running the simulation in this setup, the model neuron fired at 18.0 ± 0.3 Hz, which was an even higher frequency than exhibited by those input spike trains with the highest firing rates (Fig 7F). At the same time, the mean correlation coefficient of the strong inputs with the output spike train had doubled to 0.21 ± 0.09, with the largest values exceeding 0.4 (Fig 7E and 7F). Furthermore, also the correlation coefficients of the weak inputs with the output spike train had increased to 0.02 ± 0.03 (Fig 7E).

These results show that the larger EPSP amplitudes of strong inputs were markedly depressed during ongoing activation in our default setup (Figs 4 and 5). Notably, because the weak synaptic inputs had been only mildly depressing on average (Fig 4E), removing their

short-term plasticity mechanism should only have a small net boosting effect on their total excitatory drive. To confirm this, we additionally removed the weak inputs from the model entirely (Fig 7C and 7D) and found that this indeed had no effect on the correlation coefficients of the strong inputs (Fig 7F). Thus, after removing short-term plasticity, the computational effect of the weak inputs in maximizing information transfer of strong inputs had become redundant. In this regime, the strong synapses alone could determine the spiking properties of the model neuron.

Furthermore, the slopes of the input-output curves were markedly steeper when the short-term plasticity mechanism was removed (Fig 7G), which confirms that short-term depression of strong inputs has a divisive impact on neuronal gain [28,29], therefore broadening the neuron's responsiveness to temporal correlations in input spike trains [44].

## Discussion

We combined experimental work and computational modeling to investigate how the spiking responses of L2/3 pyramidal neurons are shaped by the complex parameter space of temporal structure within synaptic inputs, synaptic strength, and short-term plasticity.

As a first step, we mapped in detail the distributions of synaptic strength and short-term plasticity in barrel cortex L2/3. We found that short-term plasticity follows a symmetrical distribution with large variance and is mildly depressing on average. Interestingly, synaptic strength and short-term plasticity were only weakly negatively correlated across our dataset. Instead, their relationship was better captured by the simple rule that synaptic connections with EPSP amplitudes below 2 mV span the full range of depression and facilitation and exhibit no pronounced average short-term plasticity. By contrast, connections with EPSP amplitudes above 2 mV are exclusively depressing, which raises the intriguing question of what the computational role is for depression of strong synapses.

We then probed the theoretical prediction that clustering of particular synaptic properties on different postsynaptic neurons could be a mechanism for implementing filtering functions in the cortex. However, we found no experimental evidence for any pronounced clustering in L2/3 pyramidal cells.

Finally, our computational model of a L2/3 neuron suggested that the ability of the strong synaptic inputs to evoke spiking in postsynaptic cells relies predominantly on the combination of their high temporal correlation and high firing rates *in vivo* and not primarily on their synaptic strength, which should be considerably depressed during ongoing activation *in vivo*. Pairing those 'driving', co-tuned synaptic connections with strong synaptic weights, however, as has been reported for rodent V1 [6], substantially amplifies their ability to transmit information to the output spike train.

### Paired recording experiments confirm minimal stimulation results

We used minimal stimulation of axons of passage to be able to map multiple, different synaptic connections formed with the same postsynaptic neurons, which is difficult to achieve with paired recordings. To assess the technical caveats of minimal stimulation (see *Results*), including the possibility that multiple afferent axons are stimulated, that different inputs are not independent, and the unknown origin of the stimulated axon, we additionally performed whole-cell recordings of pairs of synaptically connected L2/3 pyramidal neurons. Reassuringly, we found that the distributions of EPSP amplitudes, paired-pulse ratios, and their relationship were not different between the two methods. This shows that the data we obtained with minimal stimulation are consistent with unitary synaptic connections between L2/3 pyramidal cells. Moreover, we did not observe any clustering of EPSP amplitudes or paired-pulse ratio on

the same postsynaptic neurons, which would be expected if different axons mapped with minimal stimulation were not independent of one another. This suggests that we recorded predominantly from unitary and independent connections between L2/3 pyramidal neurons and that the limitations of the minimal stimulation method did not contaminate our dataset substantially.

## Short-term plasticity in barrel cortex L2/3

While mild average depression in rodent barrel cortex L2/3 is in agreement with previous reports [4,8], other studies have found excitatory L2/3 synapses in sensory areas to be moderately facilitating on average [9,10,32]. Our finding of average short-term depression could be influenced by an elevated calcium concentration in the extracellular solution (2.5 mM) compared with the latter studies (which used around 2 mM). However, this seems unlikely given that also [4] and [8] reported mild depression using similar calcium concentrations of 2 mM.

Intriguingly, [9] and [10] used paired recordings in layer 2 (L2). While [32] conducted paired recordings across the entire thickness of L2/3, they reported only mild average facilitation with large overall heterogeneity. We recorded predominantly from neuronal somata in superficial L2/3, likely corresponding to the layer investigated by [9] and [10], but stimulated axons of passage across the entire depth of L2/3. Thus, differences between these datasets may indicate differences in synaptic properties between L2 recurrent connections [9,10] and the L3 -> L2 pathway. This is in line with a growing body of literature describing structural [45] and functional [46,47] differences between the neuronal circuits in L2 and L3 and further supports the notion that L2 and L3, which are routinely considered to constitute a single computational entity, may in fact possess different computational properties [45,47].

## No evidence for a statistical bias of synaptic innervation on L2/3 neurons

Interestingly, we found no statistical bias of synaptic strength or short-term plasticity of synaptic connections formed with the same pyramidal neurons in L2/3. Instead, our data suggest that synaptic inputs formed with a given L2/3 neuron are not markedly correlated, but that their strengths and short-term plasticity instead follow the same distribution as that of all synaptic connections across the neuropil. Such statistical biases have been hypothesized to explain lognormal firing rate distributions in cortex [35] and have been proposed as a potential mechanism for endowing neurons with high-pass or low-pass filter properties that may underlie integration and differential activation [14,38,48]. Importantly, by demonstrating the absence of a pronounced systematic biases on the single-cell level, our experimental results provide a biological constraint for theoretical models of how these particular computations may arise in L2/3.

Because we have characterized synaptic strength and short-term plasticity through somatic whole-cell recordings, we cannot exclude the intriguing possibility that statistical biases of synaptic innervation may in fact exist on the level of dendritic branches, in which case such computations may be implemented on a sub-cellular level [49–51]. Further experiments will be necessary to investigate this possibility.

## Modeling the interplay of synaptic strength, short-term plasticity, and input synchrony

To address the computational role of the pronounced short-term depression of strong connections and to investigate how synaptic strength, short-term plasticity, and temporal properties in presynaptic spike trains shape the firing properties of cortical neurons, we generated a leaky integrate-and-fire model of a L2/3 pyramidal neuron and systematically manipulated these

parameters in our simulation. The synaptic inputs to the model neuron were constrained by physiological data obtained from our own *in vitro* recordings and by *in vivo* data adopted from the literature [6,25].

In its default setup, the model was set to produce sparse firing at around 5 Hz, in agreement with the average spike rate reported for L2/3 in mouse barrel cortex [25]. Its output spike train exhibited the highest temporal correlation with the strong synaptic inputs [6]. In addition, our simulations could reproduce the effects of multiplicative gain modulation through synaptic background activity [52,53] and through short-term depression of strong synapses [28,29].

We found that synaptic background activity carried through the weak synapses contributed critically to information transfer of strong inputs by depolarizing $V_m$ and through a stochastic resonance-type effect [54]: while being incapable of evoking spiking by itself, the weak inputs enabled the model neuron to operate in a regime in which the cell became sensitive and responsive to coincident strong inputs [17,43,55–59]. Even then, the high firing rates and the synchronous activity of multiple strong synapses were needed to evoke spiking in the model neuron [16–19,60]. Notably, synaptic strength alone did not determine which presynaptic cells could evoke spikes [7].

The computational role of the relationship between short-term plasticity and synaptic strength has not been addressed in detail in studies of cortical processing. Interestingly, the pronounced short-term depression we observed for synaptic connections that elicited large EPSPs *in vitro* proved necessary to counterbalance the high firing rates, high temporal correlations, and large EPSP amplitudes of strong inputs during ongoing stimulation and was critical for maintaining the responsiveness of the postsynaptic neuron towards input spike trains with the highest temporal correlation. This suggests that short-term depression could act as one of the mechanisms that prevent runaway excitation in the recurrent L2/3 circuitry.

## A view on orientation tuning in columnar and 'salt-and-pepper' cortices

The notion that the minority of strong synaptic inputs determines the response properties of cortical neurons [6,61] has recently been challenged by apparently conflicting findings made in V1 of the ferret [7]. In mouse V1, neurons with the most similar receptive field properties *in vivo* also formed the strongest synaptic connections with each other, as assessed *in vitro* [6]. By contrast, the response selectivity of neurons in ferret V1 *in vivo* was shown to be determined by the cumulative weight of all driving synapses–weak and strong. Intriguingly, the response selectivity could not be predicted from the tuning of strong synapses alone [7]. When we shuffled the EPSP amplitudes in our simulation, we indeed found that the correlated inputs with the highest firing rates still drove the output spike train and not those inputs with the strongest synapses. However, because responsiveness was strongly reduced under this condition, other amplification mechanisms, such as the dendritic clustering of smaller synapses observed by [7] could play an important computational role.

Our observations may provide a framework to reconcile these apparently contradictory findings. In the columnar V1 of carnivores [62], presentation of simple visual stimuli activates populations of neighboring neurons within the same orientation column [63]. The axons of pyramidal cells in the superficial layers of V1 form a primary cluster of synaptic boutons around their own somata [64]. Thus, unlike in rodents, these neurons are excited by many neighboring neurons with the same orientation tuning and ocular dominance. Therefore, 'columnar' orientation maps, which are found in visual areas of higher mammals [65–68] may provide the basis for the "strength by numbers" necessary to generate tuned responses [7], without the additional requirement of stronger synapses between co-tuned neurons. Our finding that the high temporal correlation and firing rates of strong inputs, and *not* their larger

synaptic strength primarily drive spiking supports this idea and is consistent with the observation that spikes in cat V1 are phase-locked with the local field potential, which reflects synchrony within local neuronal populations [60].

By contrast, the 'salt-and-pepper' organization of rodent V1 [69], means that oriented stimuli activate a spatially diffuse network [63]. Therefore, neurons may receive fewer synaptic connections overall from similarly tuned cells and temporal correlation and firing rates alone may be insufficient to achieve orientation tuning. Our observation that pairing large EPSP amplitudes with correlated input spike trains further enhances the capacity of driving inputs to transmit information suggests that this predicted 'lack of strength by numbers' in rodent V1 may be compensated for by stronger synapses between similarly tuned neurons [6]. This, however, leads to the prediction that in mouse V1, the temporal structure in input spike trains from similarly tuned neurons also plays a key role in generating orientation tuning *in vivo*, a prediction that could be tested experimentally.

In summary, our results contribute to a nuanced framework of how cortical neurons could utilize interactions between the biophysical properties of chemical synapses, the temporal structure of input spike trains, and 'noise' in neuronal networks for efficient computation.

## Methods

### Ethics statement

Animal experiments were conducted under the license of Kevan A.C. Martin (Institute of Neuroinformatics, University of Zürich & ETH Zürich, Zürich, Switzerland). Animal handling and experimental protocols were approved by the Cantonal Veterinary Office, Zürich, Switzerland.

### Slice preparation

Cortical slices were obtained from 28 male B6/C57 mice between 21 and 28 postnatal days of age. Animals were anesthetized with isoflurane, decapitated, and their brains were removed quickly and immersed in ice-cold slicing artificial cerebrospinal fluid (ACSF, containing, in mM: 87 NaCl, 75 sucrose, 26 NaHCO3, 10 glucose, 7 MgSO4, 2.5 KCl, 1 NaH2PO4, and 0.5 CaCl2, continuously oxygenated with 95% O2, 5% CO2). Coronal slices containing the barrel cortex were cut at a thickness of 300 μm on a vibratome and transferred to a chamber containing recoding ACSF (containing, in mM: 119 NaCl, 26 NaHCO3, 10 glucose, 1.3 MgSO4, 2.5 KCl, 1.25 NaH2PO4, and 2.5 CaCl2, continuously oxygenated with 95% O2, 5% CO2). The slices were kept in recording ACSF at room temperate until the recordings.

### Electrophysiology

Patch pipettes (pipette resistance: 5–7 MΩ, pipette tip diameter: 2 μm) were pulled from borosilicate glass using a P-97 puller (Sutter Instruments) and filled with intracellular solution (containing in mM: 105 K-gluconate, 20 KCl, 10 Na-phosphocreatine, 2 Mg-ATP, 2 Na-ATP, 0.3 GTP, and 10 HEPES, the pH was set to 7.2 with KOH). Biocytin (0.5%) was added to the intracellular solution to stain the recorded neurons. Whole-cell patch-clamp recordings were obtained at 34–36°C from visually identified L2/3 neurons in barrel cortex under an Olympus BX61W1 microscope equipped with infrared differential-interference contrast optics and a 10x and a 60x water-immersion objective. Data were acquired with a Multiclamp 700A amplifier (Axon Instruments), sampled at 10 kHz, filtered at 3 kHz (Digidata 1322A, Axon Instruments), and monitored with the software *pClamp* (Molecular Devices). We did not add GABA [41,70,71] or NMDA antagonists to the bath [41,70], as previous studies have not reported any

discernible effects on the EPSP waveform by including these blockers in minimal extracellular stimulation experiments [40,72,73].

Following break-in, the access resistance was typically in the range of 15–30 MΩ and recordings with an access resistance > 30 MΩ were discarded. The bridge potential was compensated and liquid-junction potential (estimated to be 13.9 mV) was not corrected. $V_m$ after break-in ranged from -85 to -70 mV. If $V_m$ drifted during recordings, a holding current (typically < 50 pA) was injected to keep the membrane at its initial resting potential, which was rarely necessary. $V_m$ was not allowed to drift outside a range of -85 to -70 mV. Because $V_m$ was close to the reversal potential of $GABA_A$ in all experiments, we expect there was no contamination of our recorded EPSPs by inhibitory connections.

Minimal stimulation of single axons of passage was performed according to established protocols [40,41], as follows. After establishing whole-cell recordings, we identified presynaptic axons forming synapses with the recorded cells by carefully moving a monopolar extracellular stimulation electrode (filled with ACSF) through L2/3 at an oblique angle and delivering repeated 0.1 ms current pulses of 10–12 µA amplitude using an A360 stimulator (World Precision Instruments) until an EPSP was detected in the recorded neuron. Synaptic connections were typically detected when the stimulation electrode was located 20–400 µm distant from the soma of the recorded cell. To achieve stimulation of putative single axon fibers synapsing onto the recorded neuron, we then decreased the stimulation amplitude until the EPSP was not elicited anymore and subsequently increased the stimulation amplitude until the smallest observable EPSP was evoked reliably in an all-or-none manner in a fraction of trials [40,41]. The final stimulation amplitude was set to this level (typically 5–16 µA). We only recorded synaptic connections that showed little or no variability in the latency of evoked EPSPs from trial to trial. We then performed 20 ms paired-pulse stimulation at a low frequency (0.2 Hz) for at least 30 sweeps. After recordings, we carefully assessed each sweep by eye in *pClamp 9* (Molecular Devices) and included only those sweeps in the final dataset for which an EPSP was evoked following both extracellular stimulation pulses and whose evoked EPSPs were not contaminated by spontaneously occurring EPSPs. As an additional control to ensure that we were stimulating single axons of passage [41] and that the synaptic connection remained stable throughout the recording period, we only included synaptic connections if the EPSPs at the end of the recording had the identical average amplitude, latency, and shape compared to the first evoked minimal stimulation EPSPs. Our final dataset contained on average 11.2 ± 5 sweeps per synaptic connection (range of 6 to 35 sweeps).

Following the minimal stimulation protocol, we carefully moved the extracellular stimulation electrode to other locations in the L2/3 neuropil to identify different axon fibers forming synapses with the same recorded neuron. To minimize our chances of recording from the same presynaptic axon again, great care was taken not to stimulate in the same location multiple times, and synaptic connections were only included when their location of stimulation was > 50 µm away from all previous stimulation locations, as assessed in 10x overview images during recordings. At the end of each experiment, we injected current steps into each neuron to characterize its firing pattern as regular-spiking (i.e., putative excitatory/ pyramidal neuron) or fast-spiking (putative inhibitory/ interneuron).

Paired recordings were performed as described in [42] and with the same intracellular solution and ACSF as in our minimal stimulation experiments.

## Histology

After recordings, slices were immediately fixed in 15% picric acid, 4% paraformaldehyde, and 0.5% glutaraldehyde in 0.1 M phosphate buffer (PB) overnight. Fixed slices were then washed

in PB, incubated in an ascending sucrose ladder for cryoprotection, quickly frozen in liquid nitrogen, and treated in 3% hydrogen peroxide and 10% methanol in phosphate-buffered saline (PBS) to quench endogenous peroxidases. After washing in PBS and tris-buffered saline (TBS), the slices were treated with the Vectastain ABC Kit (Vector Laboratories, catalog # PK-6100, RRID: AB_2336819) in TBS at 4˚C overnight. Following washing in TBS, biocytin was visualized using nickel-diaminobenzidine (Ni-DAB) tetrahydrochloride and hydrogen peroxide treatment, followed by a series of washes in PB to terminate the reaction. Sections were then embedded in Mowiol (Sigma Aldrich) and cover slipped. Z-stacks of the recovered neurons were imaged under an Olympus BX61 microscope to cross-check the previously determined electrophysiological cell type with anatomy. Pyramidal cells were identified on the basis of their dendrite morphology (e.g., spiny dendrites) and corresponded with the previously recorded regular-spiking firing pattern.

## Analysis of electrophysiological data

We analyzed each postsynaptic potential evoked with paired-pulse stimulation individually with *Stimfit* [74] and measured its peak amplitude, coefficient of variation, onset latency (i.e., the time from the onset of the extracellular stimulation artifact to the onset of the evoked postsynaptic potential) and 10% - 90% rise time. We then averaged these respective measurements across all sweeps. The EPSP was defined as the postsynaptic potential evoked by the first pulse of the paired-pulse paradigm, i.e., before short-term plasticity took place. The paired-pulse ratio was defined as the peak amplitude of the second evoked postsynaptic potential divided by the peak amplitude of the first evoked postsynaptic potential (i.e., the EPSP). Further statistical analyses were done in *MATLAB* (MathWorks) and *Prism* (GraphPad). EPSP amplitude and paired-pulse ratios obtained with paired-recording experiments were measured in an analogous fashion.

To obtain an unbiased population distribution for a given experiment, we excluded all afferent synaptic connections formed with the postsynaptic neuron in that experiment, but otherwise included all other connections recorded in regular spiking neurons. The cell distribution for a given experiment included all afferent synaptic connections formed with the postsynaptic neuron in that experiment. The cell distributions were compared to the population distribution with the *kstest2* function in *MATLAB* (Kolmogorov-Smirnov test).

We conducted a post-hoc Monte-Carlo power analysis to estimate which effect sizes (i.e., systematic differences between mean EPSP amplitudes or mean paired-pulse ratios between the cell distribution and the population distribution) were detectable given the sample sizes in our dataset. We did this for each experiment individually by bootstrapping new cell distributions with systematically different means and then performing Kolmogorov-Smirnov tests against the population distribution.

Specifically, for the power analysis for paired-pulse ratios, we first formalized the paired-pulse ratio population distribution for each experiment as a normal distribution with the same mean and standard deviation as the experimentally observed paired-pulse ratio population distribution for that experiment. To test which effect sizes were detectable, we then formalized a range of possible underlying generator distributions for the paired-pulse ratio cell distribution for that experiment by varying the mean of the population distribution in steps of $\pm 0.1$ units. By doing so, we designed a range of generator distribution for the paired-pulse ratio cell distribution with systematically different means. For each one of these cell generator distributions, we then drew the same number of random samples that were present in the experimentally observed cell distribution (i.e., between 5 and 8) and ran a Kolmogorov-Smirnov test against a random sample drawn from the formalized population distribution (containing the same

number of entries as the population distribution for that experiment). We chose to approximate our experiments statistically with 'sampling with replacement' from a generator distribution in this manner because we could record experimentally only from a tiny fraction of the thousands of synaptic inputs formed with a L2/3 pyramidal neuron. Under these conditions, the total number of synapses formed with the neuron should be negligible.

This analysis was repeated 10,000 times for each cell generator distribution and the statistical power for detecting an effect of a certain size (i.e., the systematic difference in the means between the underlying cell generator distribution and the population distribution) was defined as the fraction of trials that yielded a significant p-value ($\alpha = 0.05$), see *Results*. The power analysis for EPSP amplitudes was done in an analogous fashion with the only exception that lognormal distributions were used instead of normal distributions, in accordance with our results.

Because our dataset contained 8 experiments for which at least 5 afferent connections were mapped, there were 8 chances for detecting a significant difference between a cell and the population distribution across our experimental series. Thus, a simple binomial model can be used to ask: which systematic difference in paired-pulse ratios should have been observed in at least one of these 8 experiments at the 95% significance level? To answer this, we computed the probability density functions for obtaining zero as a realization (i.e., the likelihood of observing no significant difference across any of the 8 experiments) of simple binomial functions with N = 8 (i.e., the number of our independent experiments) and P = the average probability of observing a given effect size in a single experiment (as derived above, see *Results*). We then repeated these analyses in an analogous fashion for the EPSP amplitude distributions.

## Modeling approach

We generated a leaky integrate-and-fire model of a L2/3 pyramidal neuron based on the passive biophysical properties of our *in vitro* recordings and published *in vivo* data. The model received inputs from 270 synaptic connections whose spike times, synaptic weights and short-term plasticity parameters were set as described in the following sections. Briefly, we first constructed 270 spike trains whose pairwise correlation coefficients and firing rates reproduced *in vivo* observations from rodent L2/3 (see *Results*). We then assigned these spike trains EPSP amplitudes and corresponding paired-pulse ratios that reproduced our *in vitro* data. Synaptic strength was then tuned such that the EPSP amplitudes in the model neuron matched exactly the somatic EPSP amplitudes we had measured *in vitro* (see below).

## Generating input spike trains with temporal correlations following *in vivo* data

We generated 270 input spike trains whose pairwise correlation coefficients matched the *in vivo* data reported by [6], i.e., the minority of (strong) input spike trains exhibited high pairwise correlation coefficients, while the remaining majority of (weak) input spike trains were progressively less correlated. We first generated a template spike train of 10 s duration that exhibited a sparse and irregular temporal structure by using an inhomogeneous Poisson renewal process and sampling inter-spike interval durations from a gamma distribution (shape k = 1.1, inter-spike interval mean of 40 ms) at 1 ms time steps, which resulted in an average firing rate of 25 Hz. We convoluted the template spike train with Gaussian envelopes of different standard deviations ($\sigma_{Gaussian}$) to generate a set of 270 new spike trains with precisely defined correlation statistics [75]. We divided the 270 inputs into strong (n = 35, i.e., 13% of inputs) and weak inputs (n = 235, i.e., 87% of inputs) based on the relationship between EPSP amplitude and short-term plasticity we had found *in vitro* (i.e., synapses with EPSP

amplitudes > 2 mV (10 / 74 synaptic connections, i.e., 13.5%) were exclusively depressing, while synapses with EPSP amplitudes < 2 mV displayed the full range of short-term plasticity). In order to set up these two populations of input spike trains with corresponding temporal correlation statistics, we sampled $\sigma_{Gaussian}$ from two uniform distributions for strong ($\sigma_{Gaussian}$ between 5 and 10 ms, n = 35) and weak synaptic inputs ($\sigma_{Gaussian}$ between 10 and 100 ms, n = 235) [75]. The resulting 270 $\sigma_{Gaussian}$ values were ranked and assigned to the 270 input spike trains. For each one of the 270 input spike trains, we convoluted the spike times of the template spike train with a Gaussian envelope whose standard deviation was set by each spike train's respective $\sigma_{Gaussian}$. By doing so, for each spike train, we obtained a 10 s time course consisting of a sum of Gaussian distributions representing the respective spike probability over time. Because of the iteratively increasing $\sigma_{Gaussian}$, this spike probability distribution for spike trains with increasing indices continuously broadens and flattens with respect to the template spike train. We then generated the discrete spike times for each input spike train by drawing spike times from these time-dependent spike probability distributions using an inhomogeneous Poisson process. The resulting 270 spike trains had continuously lower pairwise correlation coefficients with the template spike train.

Finally, we accounted for the fact that, in barrel cortex *in vivo*, correlated synaptic inputs tend to fire at higher frequencies, while uncorrelated inputs fire at lower rates [6,25]. We parametrized the lognormal firing rate distribution measured by [25] in mouse barrel cortex L2/3 *in vivo* (mean ± s.d.: 4.16 ± 8.33 Hz) and drew 270 random 'target firing rates' from it. These values were ranked and assigned to the 270 input spike trains, such that spike trains with higher pairwise correlations with the template spike train also displayed higher target firing rates. We then removed stochastically individual spikes from each input spike train such that the average firing rate of each spike train matched the respective target firing rate.

After the 270 input spike trains had been generated in this manner, we verified that their pairwise correlation coefficients [6] and firing rates [25] matched experimental data obtained in rodent L2/3 *in vivo* (see *Results*, Figs 4 and S3). This process was repeated 100 times to generate 100 different sets of spike trains to be run in the model.

## Generating EPSP amplitude and corresponding paired-pulse ratio distributions

To assign realistic EPSP amplitudes to the 270 model inputs, we parametrized the EPSP amplitude distribution we measured *in vitro* with a lognormal distribution (Fig 1D, see *Results*) and randomly drew 270 EPSP amplitude values from it. We then generated corresponding paired-pulse ratios for these 270 EPSP amplitudes by parametrized the relationship between the second pulse ($EPSP_2$) and the first pulse (i.e., the EPSP amplitude) of the paired-pulse stimulation paradigm that we had recorded *in vitro* (Fig 1C) with an exponential decay function. Critically, the jitter of the experimentally recorded $EPSP_2$ values around this fitted curve did not differ significantly from a Gaussian distribution (non-parametric Kolmogorov Smirnov test, p value of 0.48) centered around zero with a standard deviation of ± 0.192. This standard deviation captures the natural variance of the ratio of $EPSP_2$ to the EPSP amplitude and was subsequently used to generate our modeling data. For each of the selected 270 EPSP amplitudes, we first assigned a corresponding $EPSP_2$ by using the value predicted by the fitted exponential decay function for the given EPSP amplitude. We then added variance to the selected value as a number drawn from a random Gaussian process with a mean of 0 and a standard deviation of 0.192. Finally, we verified that the resulting EPSP distribution, paired-pulse ratio distribution, and their mapping corresponded to our *in vitro* recording data (see *Results*).

## Modeling short-term plasticity dynamically during presynaptic spike trains

The paired-pulse ratio captures a synapse's short-term plasticity response for two subsequent release events at a stereotypical time interval. To model short-term plasticity dynamically for ongoing activation during spike trains with variable inter-spike intervals, we formalized the short-term plasticity properties of our synapses into a general form by utilizing the widely-used extended Tsodyks-Markram model [76,77]:

$$\frac{dR(t)}{dt} = \frac{1 - R(t)}{\tau_{rec}} - u(t) \cdot R(t) \cdot \delta\left(t - t_{sp}\right) \tag{1}$$

$$\frac{du(t)}{dt} = \frac{U - u(t)}{\tau_{facil}} + f(1 - u(t)) \cdot \delta\left(t - t_{sp}\right) \tag{2}$$

Briefly, short-term depression (Eq 1) is modeled as the depletion of the synaptic vesicle pool available for release $R(t)$, with $u(t) \cdot R(t)$ following a preceding release event at time $t_{sp}$, which is counterbalanced by vesicle pool recovery at a time constant $\tau_{rec}$. Short-term facilitation (Eq 2) is modeled as an increase in release probability $u(t)$, with $f(1 - u(t))$ following a preceding spike at $t_{sp}$, which decays to the baseline release probability $U$ with a time constant $\tau_{facil}$. Thus, a continuum of synaptic depression to facilitation can be modeled by specifying the values of the parameter set $\Theta = \{\tau_{rec}, \tau_{facil}, U, f\}$ [34,78].

To do so, we derived $\Theta$ for each one of the 270 model synapses as a function of their paired-pulse ratio, as follows. A computationally optimized form of Eqs (1) and (2) was derived by [34] by integrating between spikes $n$ and $n+1$ at time $\Delta t n_n$ apart:

$$R_{n+1} = 1 - (1 - R_n(1 - u_n))e^{-\frac{\Delta t_n}{\tau_{rec}}} \tag{3}$$

$$u_{n+1} = U + \left(u_n + f(1 - u_n) - U\right)e^{-\frac{\Delta t_n}{\tau_{facil}}} \tag{4}$$

The EPSP amplitude at spike $n$ can be calculated as:

$$EPSP_n = A \cdot R_n u_n \tag{5}$$

[76], where $A$ is an adjustable weight parameter that convolves phenomenologically several physiological strength parameters, such as the number of release sites, quantal size, and cable filtering properties. The paired pulse ratio $PPR$ is the ratio of the EPSP at spike $n+1$ and the EPSP at spike $n$:

$$PPR = \frac{A \cdot R_{n+1} u_{n+1}}{A \cdot R_n u_n} \tag{6}$$

At time $t = 0$, when no preceding spike occurred, the steady-state value of $R_n = 1$ and of $u_n = U$, and Eq (6) can be simplified to:

$$PPR_0 = \frac{R_{n+1} u_{n+1}}{U} \tag{7}$$

By inserting Eqs (3) and (4) for $R_{n+1}$ and $u_{n+1}$, we can rewrite Eq (7) as

$$PPR_0 = \frac{\left(1 - Ue^{-\frac{\Delta t_n}{\tau_{rec}}}\right)\left(U + f(1-U)e^{-\frac{\Delta t_n}{\tau_{facil}}}\right)}{U} \tag{8}$$

Critically, $PPR_0$ in Eq (8) at $\Delta tn_n = 20\ ms$ (i.e., $PPR_0^{20ms}$) describes exactly our experimental paired-pulse stimulation protocol. This allowed us to obtain a parameter set $\Theta$ for each synapse as a function of its 20 ms paired-pulse ratio.

## Defining the short-term plasticity parameter set $\Theta$ for each synapse

To do so, we varied $\Theta$ on a continuum ranging from strong depression to strong facilitation according to [34] (Table 1), which resulted in a large dataset of uniquely defined $\Theta$s and corresponding $PPR_0^{20ms}$ values. For each of our 270 model synapses, we then chose the parameter set $\Theta$, whose resulting $PPR_0^{20ms}$ value matched most closely the paired-pulse ratio we had previously assigned to that synapse (see above). By obtaining a unique parameter set $\Theta$ for each synapse, we could then compute its $R_{n+1}$ and $u_{n+1}$ during continuous spike trains using Eqs (3) and (4), respectively. When starting each new run of the simulation, $R_{n+1}$ and $u_{n+1}$ were initialized to $R_n$ and $u_n$, respectively, such that the short-term plasticity parameters were reset to their initial, 'naïve' values.

## Modeling EPSP amplitude and paired-pulse ratio

Next, we converted the discrete spike times of each input spike train into a continuous synaptic conductance time course. For that, each spike was convoluted with an alpha function that was scaled by the respective input's designated synaptic strength and dynamic short-term plasticity properties:

$$g_{syn}(t) = PPR_n \cdot g_{max} \cdot \frac{(t - t_{spike})}{t_{peak}} \cdot e^{-\frac{(t - t_{spike} - t_{peak})}{t_{peak}}} \tag{9}$$

The peak conductance $g_{max}$ was fit for each input synapse such that when that synapse was activated from rest in the model ($g_{leak} = 0.01$ mS; $\tau_m = 20$ ms; as measured *in vitro*, see below), it evoked the desired EPSP amplitude. The synaptic time constant $t_{peak}$ was set to 1 ms. After fitting $g_{max}$ with this set of parameters, EPSP rise times were in excellent agreement with our *in vitro* recordings. For each spike $n$ in the input spike train, we dynamically adjusted the conductance scaling factor $PPR_n$ according to Eq (6), which is a function of that synapse's short-term plasticity parameter set $\Theta$ and its previous history of spiking. The resulting 270 conductance traces were summed into a single conductance trace containing all synaptic inputs to the model, which was then used as input to the neuron model (see below).

## Leaky integrate-and-fire model of L2/3 neuron

We implemented Euler's Method in *Python 3.7* to simulate numerically the voltage change $dV_m$ of the model using the equation:

$$dV_m = \left(-(V_m - V_{rest}) + \frac{I_{syn}}{g_{leak}}\right) \cdot \frac{dt}{\tau_m} \tag{10}$$

**Table 1. Parameter sets $\Theta$ for strongly depressing and strongly facilitation synapses, adopted from [34].**

| Synaptic short-term plasticity | $\tau_{rec}$ | $\tau_{facil}$ | U | f | $PPR_0^{20ms}$ |
|---|---|---|---|---|---|
| Strong depression | 1700ms | 20ms | 0.7 | 0.05 | 0.3 |
| Strong facilitation | 20ms | 1700ms | 0.1 | 0.11 | 1.8 |

We set the leak conductance $g_{leak}$ to 0.01 mS (corresponding to an input resistance $R_{input}$ of 100 MΩ) and $\tau_m$ to 20 ms, which were the median values recorded *in vitro* across the pyramidal neurons in our minimal stimulation experiments. When $V_m$ crossed an action potential threshold of -50 mV, $V_m$ was reset to $V_{rest}$ for a duration of 10 ms to account for the action potential time course and refractory period of the neuron. The resting membrane potential $V_{rest}$ was set to -70 mV, which resulted in an output firing rate of 5.7 Hz, in accordance with *in vivo* recordings [25]. The synaptic input current to the model cell $I_{syn}$ was dynamically computed for each simulation time step from the input conductance trace (see above) using the equation:

$$I_{syn}(t) = g_{syn}(t) \cdot (V_m(t) - E_{syn}) \tag{11}$$

The reversal potential $E_{syn}$ was set to 0 mV, i.e., the reversal potential of the AMPA synapses.

## Modeling the interplay of synaptic strength, short-term plasticity, and temporal correlation in presynaptic spike trains

After the model was set up in this manner, we simulated the voltage response of the model cell following activation of the 270 input synapses with the corresponding presynaptic spike trains. We convoluted the discrete spike times of the output spike train of the model neuron and each one of the 270 input spike trains into continuous functions with an exponential filter ($\tau$ = 10 ms) [79] and computed the pairwise Pearson's correlation coefficients between each input spike train and the output spike train. Additionally, we quantified the input-output relationship of the model neuron as the probability of spike generation as a function of the number of coincident inputs in the 20 ms time window preceding the output spike. As described in the *Results*, we then manipulated the respective population of active synapses and their synaptic parameters in the simulation to investigate the interplay of synaptic strength, short-term plasticity, and temporal correlation in presynaptic spike trains. We computed mean correlation coefficients, input-output curves, and corresponding standard deviations by repeating each simulation setup for the 100 sets of spike trains (see above).

## Robustness analysis for model parameters

We assessed whether our modeling results held true across a range of different biophysical parameters that have been reported for L2/3 pyramidal neurons *in vitro* and *in vivo*. In our initial setup, $R_{input}$ and $\tau_m$ were set according to our own *in vitro* measurements (see above) and $g_{max}$ was fit for each synapse such it produced the desired EPSP amplitude under these conditions. We kept the original $g_{max}$ value for each synapse fixed (which was fit for $R_{input}$ = 100 MΩ) and changed (1) $R_{input}$ to 80 MΩ, in agreement with *in vivo* recordings in barrel cortex L2/3 pyramidal cells [80], (2) the refractory period to 20 ms, and (3) $\tau_m$ to 10 ms, in agreement with *in vivo* recordings in barrel cortex L2/3 pyramidal cells [80]. In the latter setup, $V_{rest}$ was set to -65 mV (instead of -70 mV), such that the model neuron fired with frequencies > 2 Hz in its default setup to enable the correlation and gain analyses. Finally, we set all parameters to values reported *in vivo*: $\tau_m$ to 10 ms, $R_{input}$ to 60 MΩ, and $V_{rest}$ to -60 mV [9,13,80]. For models (3) and (4), $g_{max}$ was fit with $R_{input}$ = 100 MΩ, $\tau_m$ = 10 ms.

## Supporting information

**S1 Fig. EPSP amplitude and paired-pulse ratio measured with minimal stimulation are uncorrelated with animal age. A** EPSP amplitudes as a function of the age of the animals at

the time of the experiment. **B** 20 ms paired-pulse ratios as a function of the age of the animals at the time of the experiment. Pearson correlation coefficients are indicated.
(PDF)

**S2 Fig. Paired recordings suggest short-term plasticity remains stable during ongoing stimulation. A** Example traces of postsynaptic responses to trains of four action potentials fired in the presynaptic neuron at inter-spike intervals of 20 ms (i.e., 50 Hz; stimulation pulses indicated). Traces are averages and sorted and color-coded by decreasing EPSP amplitude (i.e., response to 1$^{st}$ pulse) and correspond to same colors in B. **B** All recorded synapses (n = 9) continue to depress during ongoing stimulation during trains of four action potentials fired in the presynaptic neuron. Data are sorted and color-coded by decreasing EPSP amplitude.
(PDF)

**S3 Fig. Mapping between synaptic strength, short-term plasticity, and correlation in input spike trains.** The relationships between parameter distributions reflects our *in vitro* data and *in vivo* data adopted from [6]. **A** EPSP distribution for 270 inputs generated from our *in vitro* recordings. **B** 20 ms paired-pulse ratio distribution for 270 inputs generated from our *in vitro* recordings. **C** Pairwise correlation coefficients for 270 inputs generated from *in vivo* data adopted from [6]. **D** Scatter plot of relationship between EPSP amplitudes and 20 ms paired-pulse ratios for the 270 inputs. **E** Scatter plot of relationship between EPSP amplitudes and pairwise correlation coefficients for the 270 inputs. **F** Scatter plot of relationship between 20 ms paired-pulse ratios and pairwise correlation coefficients for the 270 inputs.
(PDF)

**S4 Fig. Our computational results are reproducible for different combinations of model parameters.** We changed individual (A, B), or combinations of biophysical parameters (C, D) of our leaky integrate-and-fire model and repeated all analyses shown in the *Results* section. Panel layout corresponds to panels in the bottom rows of Figs 5–7. In simulations shown in C, D, V$_{rest}$ was depolarized to lift the output firing rate of the model cell $> 2$ Hz in the default setup, such that the correlation and gain analyses could be run. All pairwise comparisons between the experimental perturbations of the model and the default setup shown in the center panels (top and bottom) were statistically significant ($p < 0.05$, non-parametric Kolmogorov-Smirnov test) unless otherwise indicated. Shaded standard deviation bands were omitted for clarity, except for default simulation results in left panels. **A** Results for model with R$_{input}$ of 80 MΩ (instead of 100 MΩ), in agreement with recordings of barrel cortex L2/3 neurons *in vivo* [80]; all other parameters were left unchanged; note drop in output firing rate of model neuron (center-bottom panel). **B** Results for model with an 'un-physiologically' long refractory period of 20 ms following each spike event (instead of 10 ms). **C** Results for model with $\tau_m$ of 10 ms (instead of 20 ms), in agreement with recordings of barrel cortex L2/3 neurons *in vivo* [80]; all other parameters were left unchanged; V$_{rest}$ was depolarized by 5 mV [9] to allow for sufficiently high output firing rate. **D** Results for an '*in vivo*-like' model with R$_{input}$ of 60 MΩ (instead of 100 MΩ), $\tau_m$ of 10 ms (instead of 20 ms), and V$_{rest}$ of -60mV, in agreement with recordings of barrel cortex L2/3 neurons *in vivo* [9,13,80].
(PDF)

**S1 Data. Source data for Fig 1.**
(XLSX)

**S2 Data. Source data for Fig 2.**
(XLSX)

**S3 Data. Source data for Fig 3.**
(XLSX)

**S4 Data. Source data for Fig 4.**
(XLSX)

**S5 Data. Source data for Fig 5.**
(XLSX)

**S6 Data. Source data for Fig 6.**
(XLSX)

**S7 Data. Source data for Fig 7.**
(XLSX)

## Acknowledgments

We would like to thank Kevan A.C. Martin for his inspiration, support, comments on the manuscript, and funding. We would like to thank Qendresa Parduzzi for help with developing the model. As members of the Institute of Neuroinformatics, the authors are signatories of the Basel Declaration.

## Author Contributions

**Conceptualization:** Benjamin Ehret, Gregor F. P. Schuhknecht.

**Data curation:** Moritz O. Buchholz, Gregor F. P. Schuhknecht.

**Formal analysis:** Moritz O. Buchholz, Alexandra Gastone Guilabert, Gregor F. P. Schuhknecht.

**Investigation:** Moritz O. Buchholz, Alexandra Gastone Guilabert, Gregor F. P. Schuhknecht.

**Methodology:** Alexandra Gastone Guilabert, Benjamin Ehret, Gregor F. P. Schuhknecht.

**Software:** Alexandra Gastone Guilabert, Benjamin Ehret, Gregor F. P. Schuhknecht.

**Supervision:** Benjamin Ehret, Gregor F. P. Schuhknecht.

**Visualization:** Gregor F. P. Schuhknecht.

**Writing – original draft:** Gregor F. P. Schuhknecht.

**Writing – review & editing:** Moritz O. Buchholz, Gregor F. P. Schuhknecht.

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
