## [Decision Letter · Decision Letter 0]

15 Nov 2022

Dear Dr. Schuhknecht,

Thank you very much for submitting your manuscript "How synaptic strength, short-term plasticity, and input synchrony contribute to neuronal spike output" for consideration at PLOS Computational Biology.

As with all papers reviewed by the journal, your manuscript was reviewed by members of the editorial board and by several independent reviewers. In light of the reviews (below this email), we would like to invite the resubmission of a significantly-revised version that takes into account the reviewers' comments.

We cannot make any decision about publication until we have seen the revised manuscript and your response to the reviewers' comments. Your revised manuscript is also likely to be sent to reviewers for further evaluation.

Sincerely,

Boris S. Gutkin

Academic Editor

PLOS Computational Biology

Daniele Marinazzo

Section Editor

PLOS Computational Biology

Reviewer's Responses to Questions

**Comments to the Authors: **

Reviewer #1: NOTE: We reviewed this manuscript for eLife and then for JNeurosci. Although our feedback changed between the eLife and the JNeurosci manuscripts, it appears that the present PLoS CB manuscript is virtually identical to the JNeurosci manuscript: figures, line numbers, etc appear to be exactly the same. We are therefore providing essentially the same feedback here as we did for the JNeurosci manuscript.

Please accept our apologies in advance if we forgot to remove points that were already addressed, in which case we would ask for a clarification as to how the point was addressed. We have seen so many versions of this manuscript now that it is easy to get confused.

Using a combination of acute-slice patch-clamp recordings, extracellular stimulation, and computer modelling, Guilabert et al explore how a neuron's spiking output is determined by synaptic strength, short-term plasticity, and spiking structure of its inputs. In contrast to the commonly held view that synaptic strength determines spiking output, the authors argue for a more nuanced view. In particular, the authors argue that they find that "the ability of strong inputs to evoke spiking critically depended on their high temporal synchrony and high firing rates observed in vivo and on synaptic background activity," meaning synaptic strength is just one factor among many that in practice that determine a neuron's spiking output.

This paper is well written and well argued. The figures were visually pleasing and straightforward to interpret. This made for a very pleasant and enjoyable read. The computational modeling revealed interesting findings that contribute to our understanding of how the integration of synaptic inputs contributes to cortical neuron output. 

However, some claims seem a bit excessive, several of the modelling outcomes are presented as findings or evidence of realism when they seem to be a consequence of the parameter settings or alternatively not representative comparisons, the use of extracellular stimulation comes with limitations, and it is unclear why fast-spiking cells were included. Finally, the novelty is unclear, although this might be a matter of providing clarification. The study adds nuance to our thinking about what determines spiking output, even if we presumably still think synaptic weight mostly determines spiking output.

This manuscript should be highly suitable for publication in PLoS CB once these points have been addressed in full, so if the authors could address our concerns, we would strongly support publication.

MAJOR POINTS:

1. The authors make some strong claims from their data -- this could be clarified and/or toned down. For example, on p7, the authors say "these are important experimental results that contradict the theory-inspired hypothesis that synaptic inputs onto single cortical neurons may be statistically correlated", but it only seems to me that their data is "not consistent with" rather than "contradict", because I think of many caveats associated with the authors experiments (Major Points 2 and 3) and analysis (Major Point 5) that could also explain this mismatch. In general, it would seem appropriate of the authors to clearly outline the shortcomings and pitfalls of their own data and analysis.

2. Extracellular stimulation is possibly an outdated method for studying monosynaptic connections, which leads to alternative interpretations of the present study. There are several issues associated with this methodology that the authors never control for or otherwise address. This should at least be discussed.

2a. How do authors know that inputs are independent? Then mention that stimulation locations are >50µm apart, but axonal arbours are considerably larger than this. In the older plasticity literature that relied on dual electrodes to show input specificity, independence of different inputs was verified with two electrodes activated in quick succession to show absence of short-term plasticity. Did the authors do any verification of that kind, or do we just have to take their word for it, that 50-µm spacing provides independence of inputs? 

2b. PPR should be affected by extrac stim under some circumstances. With minimal stimulation, afferents can easily drop out, but when they do, they can differentially drop out on the first or the second pulse of a paired-pulse stimulation, which will apparently affect the paired-pulse ratio. Given that they authors report a correlation between the EPSP amplitude and the PPR, they need to account for the possibility that this is a consequence of using extrac stimulation rather than paired recordings. 

2c. Was excitation ever contaminated by inhibition? If no, how do the authors know that excitation was not contaminated by inhibition? Did they wash in GABA blockers to show that inhibition was never recruited, or only recruited so often? Did the authors carry out any other control?

2d. Anybody who has ever used extrac stim knows that the CV of responses decreases with increasing stim strength, as afferents are averaged and noise thereby reduced. Do the authors argue that these are unitary inputs, or a mix? If the authors wish to argue that minimal stimulation only recruits unitary inputs (like with a paired recording), then they need to report controls to that effect. If the authors do not wish to argue that recordings are unitary, then they are dealing with a mixed data set, where some recordings may be unitary, but some are not, which would impact the heterogeneity of the data set. This could for example help to explain why the PPR of large (>2mV) inputs show a smaller PPR variance, since they are the consequence of multiple inputs being averaged, leading to smaller CV. Larger PPR spread of small inputs could be due to more noise -- since the PPR is a ratio, it is sensitive to the pulse-1 amplitude in the denominator.

2e. How do the authors really know the identity of the presynaptic cell? With paired recordings, we'd ask for morphometry, spike pattern, layer localization, and/or genetic markers of the presynaptic cell. Here we just have to believe that the stim electrode location tells all we need to know. Maybe the heterogeneity of inputs is a consequence of activating different kinds of inputs?

3. Paired pulse ratio is not the same thing as the full short-term dynamics of a synapse. It is both possible and quite common for a synapse to exhibit short-term facilitation in a paired-pulse sense while still showing short-term depression overall for the remainder of a high-frequency burst. In fact, already the Tsodyks-Markram model famously shows how assessing short-term plasticity merely from the first two pulses can be misleading. The authors need to address this measurement problem.

4. It is unclear why fast-spiking cells were included in this study. The authors report that excitatory synaptic connections formed with regular-spiking L2/3 neurons do not exhibit a systematic clustering of EPSP amplitude and short-term plasticity. Did they find the same results for the excitatory connections onto fast-spiking cells? Those results should be reported too. The authors show data for EPSPs onto fast-spiking neurons in Fig 1, but then they never mention fast-spiking neurons again and do not provide any explanation as to why they did not pursue any further investigation into the integration of inputs onto fast-spiking neurons. Are those cells ever mixed up with the pyramidal cell recordings? That would seem inappropriate. I am thinking that it might be a good idea to simply remove the fast-spiking cells from this study.

5. The statistical treatment to show that there is no clustering is at times unclear. 

- First of all, Fig 2C shows how the cell 1 data is included in the population data for the comparison between cell 1 and the population, which clearly violates the requirement that the data sets be independent. The same holds true for the cell 2 - 8 comparisons. In the Methods on p22, HOWEVER, something else is stated: "To obtain an unbiased population distribution for a given experiment, we excluded all afferent synaptic connections formed with the postsynaptic neuron in that experiment, but otherwise included all other connections recorded in regular spiking neurons." Please clarify this in Fig 2. Maybe we are just misunderstanding what was done and it is all good?

- Second, the power analysis relies on two approaches, but neither of them require information about the overall number of inputs that exist onto a L2 PC. Please justify this approach. Cerebellar granule cells, for example, famously have only a handful of inputs. If the authors had been recording from granule cells, they would thus be able to sample a much, much larger fraction of the overall input 'connectome' onto those cells than they now can with L2 PCs. According to Braitenberg and Schuz's many papers, your typical pyramidal cell should have something like 5000 to 10000 inputs. Presumably the power analysis of the clustering must depend on how well the overall input space is sampled, yet this parameter never enters the picture. Please clarify.

- Third, PPR depends on age over P14-P30 (e.g. Reyes & Sakmann 1999), and the authors explore P22-P29. Does PPR correlate with age in the authors' data set? Could this affect the clustering test?

- Fourth, repeated comparisons are done in Fig 2, so how are post-hoc corrections for multiple comparisons done? If you keep testing, eventually a comparison will come up significant. Page 22 says "Because our dataset contained 8 experiments for which at least 5 afferent connections...", which seems to claim that this took care of this issue, but does it really? Please clarify.

6. Several of the modelling outcomes are presented as findings or evidence of realism when they are simply a consequence of the parameter settings or alternatively not representative comparisons.

6a. On p8, the authors state "The mapping between EPSP amplitude and short-term plasticity across the model inputs (Fig. 4 A, B) followed the same relationship as observed in vitro". It would be very strange if this was not the case, since this is how and why the model was constructed. Or am I missing something here?

6b. On p10, the authors say "Critically, without further tuning, the model neuron reproduced key properties of rodent L2/3 pyramidal neurons in vivo", but R_input, tau_membrane etc span huge ranges in the acute slice, it is one of those famous long-tailed distributions, so that the authors happen to get a good match is just luck -- it cannot possibly mean anything. To convince themselves that this is pure luck, the authors should carry out a robustness analysis with respect to key parameters of the model (axial resistance, R_i, gleak at dendrite and soma, etc) -- this will show how much the firing rate can actually vary. The authors can for example start with picking an axial resistance of 70 Ohm-cm instead of 100 Ohm-cm (p23), or tweak R_i by 5% away from 1 Ohm-cm, or why not nudge the gleak of the soma something other than the arbitrarily precise selection of 0.0379 mS/cm2. Besides, many cited parameter values come from in-vitro measurements, but e.g. R_input and tau_m are famously totally different in vivo, due to background activity, so the fact that modelling and in-vivo frequencies magically match is actually evidence that they do not match. 

6c. On p11, the authors say "The average membrane voltage (Vm) of the model neuron was -65.93 mV ± 7.82 mV (Fig. 5 B, D), comparable to in vivo whole-cell recordings in mouse L2", but in-vivo recordings are done during synaptic bombardment, under unclear up/down state, and anaesthesia. While it is good to have realism, I am not sure how much we can learn from this.

6d. On p11, the authors say that when they removed the weak inputs, "The output firing rate of the model neuron dropped to 1.26 ± 0.34 Hz (Fig. 5 F) and its average Vm was hyperpolarized to -68.39mV ± 4.57mV." The authors now need to compensate for this change in V_m and R_input due to removing the weak inputs by adjusting the model accordingly, otherwise the findings are trivial -- there was just an indirect change in model parameters. Correct me if I am wrong, but I cannot see that the authors carry out such a compensation. The cited Destexhe papers have a method for such a compensatory change, for example. But it would seem simple to depolarize the cell accordingly to compensate for loss of average conductance drive, to enable comparison. Otherwise it would seem odd to claim that "Notably, synaptic strength alone did not determine which presynaptic cells could evoke spikes (Scholl et al., 2020)" (page 17).

6e. P17, "such that it was not necessary to include inhibitory synapses in the simulation to balance excitation." and then "Reassuringly, without further parameter tuning, the model exhibited key computational properties of cortical neurons that have been characterized in experiments and simulations before: the model cell produced sparse firing at around 5 Hz, which is in excellent agreement with the average spike rate reported for mouse barrel cortex L2/3" Honestly, doesn't it come across as super weird that model firing rates match in-vivo firing rates if the model doesn't even have inhibition? This line of reasoning seems odd to me.

6f. P27, "Reassuringly, we found that the resulting somatic EPSP amplitude distribution and paired-pulse ratio distribution exactly matched the target distributions we had generated". I don't get it. Why wouldn't it match? That whole paragraph starting with "By deriving paired-pulse ratios..." is not reassuring to me, it confuses me.

7. Calcium has a major impact on short-term plasticity. The authors use 2.5 mM, but make comparisons with papers that use other concentrations without mentioning this (e.g. Lefort and Petersen use 2 mM, see p16). This caveat should be made clear.

8. The novelty should be brought out better. The study adds nuance to our thinking about what determines spiking output, but it does not really change the way we think. Presumably, we still think synaptic weight mostly determines spiking output, no? Please clarify.

MINOR POINTS:

7. p4, how was the 10-90% rise time measured? From the average sweep, or from the individual sweeps and then averaged? If the former, this will due to temporal jitter tend to make the rise time longer. Please clarify.

8. p2, "interval of 20 ms", it seems more meaningful to report the frequency.

10. p4, "a scatter plot of the response amplitudes to the 2 stimulation pulse against the response amplitudes to the 1st stimulation pulse..." This is also true for individual sweeps (at least for paired recordings between pyramidal cells), because the readily-releasable pool depletes less if there is failure of release on the first pulse, so that the readily-releasable pool is closer to full for the second response, and vice versa. A corollary seems to be that lower PPR for large EPSPs may be a consequence of p_release being close to maxing out for strong synapses. By analyzing the individual sweeps, the authors may be able to show this: weak inputs may have a strong correlation between pulses 1 and 2, whereas strong inputs may not, due to maxing out p_release, thus leaving little or no space for pulses 1 and 2 to correlate.

11. The authors are often using the word "significant", which is confusing. In science, findings are there if they are significant, and not there if they are not significant, so there is no need to say that e.g. (p4) "There was no significant correlation in our data set", it is enough to say "There was no correlation in our data set". The reason this is important is because sometimes significant just means "a lot", but on p5, with "connections above 2 mV had a significantly lower mean paired-pulse ratio", I can't tell if the authors mean "a lot lower" or "statistically significantly lower". Therefore, I recommend just staying away from the s-word, "significant", unless there is some sort of special reason for using it. To denote "a lot", the authors could say "considerably" or "markedly" etc.

12. Fig. 1C, F: I would like to see the p-values for those correlations in the graph or in the figure caption.

13. p5, "To investigate this further, " should probably start a new paragraph.

14. <partially address> I do not quite understand how these panels related to the figure caption just below. Could the authors explain in the caption what these results mean, please?

15. Query, p13-14, Was short-term plasticity initial conditions run until it was relaxed, or did it start naively for each simulation run? Please clarify.

16. P16, unclear what "entire L2/3" means here.

17. P17, "the intriguing possibility that statistical biases of synaptic innervation may exist on the level of dendritic branches", please cite relevant papers. I think David DiGregorio had a paper back in 2012 showing that Purkinje Cells dendritic filtering affects PPR, I imagine similar papers must exist for neocortical pyramidal cells. Maybe try Stephen Williams and/or Greg Stuart.

18. p23, Where are the synapse located along this model dendrite? Does the synapse location affect the PPR (it would if the dendrite has e.g. I_h, like cortical pyramidal cells do)

19. P18, paragraph starting "Our result that spiking in the model neuron was driven...", How would the computer model be different if based on Scholl et al 2020 findings? I mean, does the line of reasoning presented in this paragraph really explain the difference? I smell a non-sequitur here, but maybe I am missing something?

20. P20, please report the liquid junction potential, even if not corrected for.

21. P20, "If the membrane potential drifted during recordings, a holding current was injected to keep the membrane at its initial resting potential, which was rarely necessary." Please state how much. Please clarify: Was V_m allowed to drift outside -85 to -70 mV?

24. P21, vague minimal stim criteria, "in an all-or-none manner" and "which showed little or no variability in EPSP latency or shape". Please provide specific numerical values.

26. P22, "to identify different axon fibers" and "Great care was taken not to record from the same" -- How do you know it is different, really? See Major Point 2.

27. P23, "We convolved the template spike train with Gaussian envelopes..." Isn't this vernacular use of the word "convolve"? Two instances.

29. P24, a thought: Increasing the rate of a Poisson process also lowers its CV and would tend to artificially enhance cross-correlations. How was this accounted for?

Reviewer #2: This paper combines experimental work on short term synaptic dynamics and single cell modelling to investigate synaptic integration in L2/3 cells.

After measuring synaptic dynamics, they use a 2 compartment model to measure synaptic integration. A very specific pattern of activity is used where strong synapses receive higher rate have less jitter and are more strongly correlated. (Little attention is given to higher order correlation in the inputs that will become dominant at some point).

Unsurprisingly in this setup, the many background inputs lift the membrane potential of the cell and boost the effect of the strong inputs.

The net effect of STP is weak. While the effect of the background inputs becomes weaker this likely depends on precise model parameters.

The paper is clear, but the conclusions are not particular surprising from experimental or computational point, novel, or solid. Namely, synchronous, high rate inputs are effective in driving the neuron, and a background input helps to lift it above threshold.

Because the biochemical milieu can be quite different in in vivo and in vitro (e.g. adenosine to name one), and hence dynamical synaptic properties can vary a lot, the extrapolation from in vivo to in vitro without further verification is rather risky.

Major:

The emergence of a correlation only when RS and FS cells are pooled, but not in the individual FS or RS population by itself is troublesome (Fig 1F). It suggests that an confounding explanation is that FS cells have large, depressing synapses and RS cells small, less depressing synapses.

Can this be ruled out? If not, how would that affect the result?

It would be of interest to know whether a simple linear integration model would already predict already all the observed effects. I would assume so. Currently the model is too complicated to understand mathematically, but too simple to be a good model of a neuron.

Minor:

l135. I don't see how the question of correlation/mapping is dependent on the clause ("Given their...").

I wonder why in the model Ri in the soma was set to 1 Ohm cm. It really should not matter given the large soma which should be close to electrotonically compact. (If it does there is also certainly something wrong).

Also, if the membrane capacitance was enhanced to compensate for the presence of spines, why was the somatic conductance boosted even more?

I was also surprised by the choice of making all synapses equidistant. This will increase saturation and makes one wonder whether a spatially extended model was used at all.

l778 The statement that no robustness analysis is needed, because synaptic strenghts were tuned to data is misleading and very wrong. While the indivual synaptic strengths might have matched, properties such as synaptic interaction and filtering will dependent on the precise parameter combination. If the statement were true there would be no need for detailed models at all.

l830 I presume the sd has also a factor 10^-6.

**Have the authors made all data and (if applicable) computational code underlying the findings in their manuscript fully available?**

Reviewer #1: **No: **Authors state that "Code will be deposited on GitHub and data will be made available upon request." so it was not done yet.

Reviewer #2: Yes

PLOS authors have the option to publish the peer review history of their article (what does this mean?). If published, this will include your full peer review and any attached files.

Reviewer #1: No

Reviewer #2: No
---

## [Decision Letter · Decision Letter 1]

24 Mar 2023

Dear Dr. Schuhknecht,

We are pleased to inform you that your manuscript 'How synaptic strength, short-term plasticity, and input synchrony contribute to neuronal spike output' has been provisionally accepted for publication in PLOS Computational Biology.

Best regards,

Boris S. Gutkin

Academic Editor

PLOS Computational Biology

Daniele Marinazzo

Section Editor

PLOS Computational Biology

Reviewer's Responses to Questions

**Comments to the Authors:**

Reviewer #1: The authors addressed all of our concerns in detail. This manuscript is ready for publication.

**Have the authors made all data and (if applicable) computational code underlying the findings in their manuscript fully available?**

Reviewer #1: **No: **E.g., submission says "Code will be deposited on GitHub", not "has been deposited"

PLOS authors have the option to publish the peer review history of their article (what does this mean?). If published, this will include your full peer review and any attached files.

Reviewer #1: No

---

## [Editor Report · Acceptance letter]

13 Apr 2023

PCOMPBIOL-D-22-01408R1 

How synaptic strength, short-term plasticity, and input synchrony contribute to neuronal spike output

Dear Dr Schuhknecht,

I am pleased to inform you that your manuscript has been formally accepted for publication in PLOS Computational Biology. Your manuscript is now with our production department and you will be notified of the publication date in due course.

With kind regards,

Zsofi Zombor
